# HFE-Related Hemochromatosis May Be a Primary Kupffer Cell Disease

**DOI:** 10.3390/biomedicines13030683

**Published:** 2025-03-10

**Authors:** Elias Kouroumalis, Ioannis Tsomidis, Argyro Voumvouraki

**Affiliations:** 1Department of Gastroenterology, PAGNI University Hospital, University of Crete Medical School, 71500 Heraklion, Greece; 2Laboratory of Gastroenterology and Hepatology, University of Crete Medical School, 71500 Heraklion, Greece; itsomidi@gmail.com; 31st Department of Internal Medicine, AHEPA University Hospital, 54621 Thessaloniki, Greece; iro_voum@yahoo.gr

**Keywords:** iron, hepcidin, ferroportin, hereditary hemochromatosis, bone morphogenetic proteins, erythroferrone, Kupffer cells, liver sinusoidal endothelial cells

## Abstract

Iron overload can lead to increased deposition of iron and cause organ damage in the liver, the pancreas, the heart and the synovium. Iron overload disorders are due to either genetic or acquired abnormalities such as excess transfusions or chronic liver diseases. The most common genetic disease of iron deposition is classic hemochromatosis (HH) type 1, which is caused by mutations of *HFE*. Other rare forms of HH include type 2A with mutations at the gene *hemojuvelin* or type 2B with mutations in *HAMP* that encodes hepcidin. HH type 3, is caused by mutations of the gene that encodes transferrin receptor 2. Mutations of *SLC40A1* which encodes ferroportin cause either HH type 4A or HH type 4B. In the present review, an overview of iron metabolism including absorption by enterocytes and regulation of iron by macrophages, liver sinusoidal endothelial cells (LSECs) and hepatocyte production of hepcidin is presented. Hereditary Hemochromatosis and the current pathogenetic model are analyzed. Finally, a new hypothesis based on published data was suggested. The Kupffer cell is the primary defect in *HFE* hemochromatosis (and possibly in types 2 and 3), while the hepcidin-relative deficiency, which is the common underlying abnormality in the three types of HH, is a secondary consequence.

## 1. Introduction

Iron is a critical element in many important biochemical pathways, implicating mitochondria, enzymatic reactions and DNA synthesis. Probably the most important role is its vital participation in the synthesis and function of hemoglobin and myoglobin. Every erythrocyte (RBC) contains more than 1.1 billion atoms thus providing more than a billion oxygen binding sites per erythrocyte. Iron homeostasis must be strictly regulated in the human body [1,2,3].

Iron overload and iron deficiency are equally detrimental. Iron overload is either genetic mostly due to hereditary hemochromatosis (HH), caused by mutations in genes implicated in iron sensing and modulation (such as *HFE*), and β-thalassemia or to acquired conditions such as chronic liver diseases [4,5].

RBCs also contain significant amounts of non-heme iron that must be removed by the only known iron exporter ferroportin (FPN1) to maintain normal RBC functions. Dysregulation of this efflux pathway promotes oxidative stress and enhanced hemolysis [1].

Clearance of aging or damaged RBCs is an important process that is fundamental in the turnover of the iron pool. This is achieved by phagocytosis of senescent and damaged RBCs by macrophage populations such as Kupffer cells of the liver and splenic macrophages. Not all signaling pathways involved in the functions of iron-recycling macrophages have been clarified. New insights emerge into the effects of iron in macrophage immune polarization [6,7].

Hemochromatosis is an iron overload syndrome characterized by normal iron-driven erythropoiesis and harmful deposition of iron in hepatocytes, heart and endocrine glands. It is caused by mutations that affect the proteins that regulate iron absorption, transport, storage and efflux. In mice, deletion of the iron regulatory protein hepcidin and genes that regulate iron biology, such as *Hfe*, transferrin receptor 2 (*Tfr2*), hemojuvelin (*Hjv*) and ferroportin (*Fpn*) cause iron overload but not organ disease. Additional co-morbidities or combined mutations may be required for organ damage [8,9].

This review will cover all aspects of iron metabolism and the role of different types of cells that are participating in the control of iron homeostasis. All the above multiple facets of iron metabolism will be covered including the molecular mechanisms implicated in RBC phagocytosis. The cross talk between macrophages LSECs and hepatocytes will also be presented as the basis for understanding the pathophysiology of HH. Moreover, the biological and epidemiological background of hemochromatosis will be analyzed and the current pathophysiology will be examined. Finally, a new hypothesis will be proposed. According to this, *HFE* HH is due to a primary defect of Kupffer cells. In all three types of HH, hepcidin-relative deficiency is a common abnormal defect considered as the primary abnormality leading to increased iron deposition. In the new hypothesis, this is a secondary phenomenon and not the primary defect at least in HFE-related HH.

## 2. An Overview of Iron Regulation

Approximately, 2 × 10^15^ iron atoms per second are required to accommodate the production of almost 200 billion red blood cells (RBCs) per day under normal circumstances in humans. Most iron in the human body is either found in hemoglobin or deposited in ferritin in hepatocytes and macrophages. Only 2–4 mg of iron is normally circulating in the plasma transported by transferrin (Tf). Iron export into plasma comes either from aged red blood cells phagocytosed by macrophages, mostly Kupffer cells, or from absorption by enterocytes. Iron from ferritin stores may also contribute to plasma iron [1,10].

Studies have shown that approximately 400 human genes code for iron proteins which take up iron directly or indirectly through heme and iron–sulfur (Fe–S) clusters [11].

### 2.1. Duodenal Absorption

Iron in the diet is oxidized to ferric iron (Fe^+3^). Ferrous iron (Fe^+2^) is more toxic because it transforms the constantly generated mild reactive oxygen species (ROS) such as superoxide and hydrogen peroxide into the highly toxic hydroxyl radicals via the Fenton and Haber–Weiss reactions that lead to serious cell damage [12].

Before absorption, dietary iron is reduced in the duodenal lumen to Fe^+2^ by the ferric reductase duodenal cytochrome b (DcytB) and then is transported through the apical membrane of the duodenal enterocytes by the divalent metal transporter 1 (DMT1). The imported iron is either stored in ferritin or it is transported to the iron exporter ferroportin (FPN1) localized in the basolateral membrane of the enterocyte. The chaperons poly (rC) binding proteins 1 and 2 (PCBP1 and PCBP2) are responsible for the transport of the iron to ferritin (FT) [13,14]. It was recently found that PCBP1 also regulates both iron absorption and iron export as a reduction of PCBP1 in the enterocyte leads to an increase in intestinal iron absorption, despite decreased FPN1 and increased serum hepcidin levels [15,16].

DMT1 (gene *Slc11a2*) is the first iron transporter and the only mechanism of non-heme iron absorption in the enterocyte. An acidic pH is the optimum for its function. The H^+^ gradient that drives DMT1-mediated iron uptake is maintained by the enterocyte brush border Na^+^/H^+^ exchanger 3 (NHE3) [17]. FPN1 is the second iron transporter.

On the other hand, heme–iron is also directly absorbed from the enterocyte through a carrier protein, possibly the heme carrier protein 1 (HCP-1) [18,19]. Heme is extremely toxic for the cell. Therefore, an effective mechanism for heme detoxification has been developed. The transcriptional repressor Btb and Cnc homology 1 (Bach1) is the initiator of this protective mechanism. Bach1 acts as a sensor for heme due to its high affinity for heme. Increasing cellular heme levels induce Bach1 degradation, initiating two protective pathways. First, the inhibitory binding of Nuclear Factor Erythroid 2-like (Nrf2) to antioxidant-response elements (ARE) for genes containing AREs [20], such as heme-regulated gene-1 *(HRG1*), and heme oxygenase (HO-1) is removed [21]. When heme levels are low, Kelch-like ECH-associated protein 1 (Keap1) maintains Nrf2 in the cytoplasm. Increased heme dissociates the Keap1-Nrf2 complex and Nrf2 is translocated to the nucleus [22,23]. Both HO-1 isoenzymes cut the protoporphyrin ring of heme yielding bilirubin, carbon monoxide and iron which is either stored within ferritin, or exported to the extracellular space via FPN1 [24]. The second consequence of the degradation of Bach1 is the production of the transcription factor SPI-C in the splenic red pulp macrophages independently of Nrf2 [25]. It should be noted that HO-1 is cytoprotective against cell death, including necrosis, necroptosis and pyroptosis. In ferroptosis, HO-1 may be detrimental as it enhances iron release. The final effect of HO-1 depends on the cells and tissues involved. HO-1 has also been involved in the regulation of autophagy [26].

Extensive reviews of heme trafficking have been recently published [27,28,29]. Figure 1 depicts the iron absorption by the enterocyte.

### 2.2. Storage

Ferritin is the most important iron deposit in cells, but it can also circulate in plasma. Serum ferritin is mostly derived from macrophages [31]. Ferritin may contain up to 4500 iron atoms in a complex structure comprised of 24 chains, both heavy (H) with ferroxidase activity, and light (L) chains [32,33].

The synthesis of ferritin chains is regulated by the iron response element (IRE)/iron regulatory protein (IRP) system (see below). The ferritin mRNAs have a single IRE in the 5′ UTR. In iron deficiency, the IRP binds to this IRE and reduces translation. When intracellular iron increases, the brake is removed upregulating ferritin synthesis [34]. In addition to cytoplasmic ferritin, iron is also stored within other cellular compartments such as nuclear ferritin (NuFt) and mitochondrial ferritin (MtFt) or even forms aggregates of hemosiderin [35]. Storage in ferritin demands oxidation of ferrous to ferric iron, which is achieved by the ferroxidase activity of H-ferritin [32].

### 2.3. Cellular Uptake and Traffic

Iron is transported to cells either bound to proteins of the transferrin family such as transferrin (Tf), melanotransferrin (MTf) and lactoferrin (Lf) or through circulating ferritin, lipocalin 2 (LCN2) and integrated in heme proteins, such as hemoglobin [35].

#### 2.3.1. Transferrin (Tf) and Transferrin Receptor (TfR) in Iron Homeostasis

Transferrin is the major iron-binding glycoprotein functioning as an iron transporter and also as a signaling regulator. Tf transports iron from every source, whether dietary or from macrophage recycling [36]. A conserved gene duplication resulted in the bi-lobed structure of Tf (lobes N and C), with each lobe transporting one iron molecule. As a result of this configuration, Tf may circulate as diferric (holoTf), monoferric N lobe, monoferric C lobe, and apotransferrin [37]. The dual function of transferrin either as an iron transporter or as a regulator ligand depends on the different affinity of iron binding of the N and C lobes leading to the differential interaction of monoferric transferrin with the two transferrin receptors [38].

Tf receptors one (TfR1) and two (TfR2) have different characteristics. The major non-redundant function of hepatocyte TfR1 is to interact with HFE to modulate hepcidin contributing to hepcidin suppression and iron overload in β-thalassemia [39].

Holo-Tf binds to TfR1, and is transported within the cells into endosomes. Proton pumps create an acidic environment inside the endosomes that dissociates the ferric iron from the Tf/TfR complex. The Six-transmembrane epithelial antigen of prostate 3 (Steap3) reduces ferric iron into the ferrous form, while DMT1 transports iron into the cytoplasm, while the Tf/TfR1 complex is recycled assisted by the sorting nexin 3 (SNX3) protein [40,41].

In addition, holoTf increases the phosphorylation of the Extracellular Signal-Regulated Kinase (ERK1/2) after attachment to the HFE–TfR2 complex, inducing the pro-protein convertase furin expression. Furin in turn intensifies the maturation of hepcidin and BMPs by cleaving their non-mature forms. A positive loop is created, increasing hepcidin expression [42,43,44].

Ferritin binds also to TfR1. However, in humans only the H-subunit of ferritin can interact with TfR1. A certain level of TfR1 expression is required for cells such as erythroblasts to integrate ferritin through TfR1 [44]. Apart from the Tf/TfR1 pathway, cells can also take up iron from non-transferrin-bound iron (NTBI), a heterogenous mixture that includes high molecular iron aggregates. NTBI is not present in healthy people, but it becomes detectable when transferrin saturations exceed 70% in diseases such as hereditary hemochromatosis and β-thalassemia major [45]. NTBI is taken up by carriers such as DMT1 or the zinc transporters Zrt–Irt-like Protein 8 (ZIP8) and Zrt–Irt-like Protein 14 (ZIP14) [40,41,46,47].

Other iron uptake pathways are operative such as the uptake of hemoglobin (Hb)–haptoglobin, heme–hemopexin, heme, and lipocalin 2, via CD163, CD91, FLVCR2, SLC22A17, Scara5, and Tim2 receptors, respectively. Once in the cytoplasm, iron enters the labile iron pool (LIP), and is utilized by mitochondria for the synthesis of heme and iron–sulfur clusters. Mitoferrins (Mnf 1/2) import iron in mitochondria and FLVCR1B export heme from mitochondria. The unused LIP is either stored, or exported out of the cell [2,35].

Additional forms of protein-bound iron are taken up by receptor-mediated endocytosis. In mice, L-ferritin and H-ferritin enter lysosomes through the Scavenger receptor class A member 5 (Scara5) and T Cell Immunoglobulin and Mucin Domain Containing 2 (Tim2) receptors, respectively [48,49]. The Tim2 gene is absent in human cells and H-ferritin is taken up by TfR1 [50].

The intracellular trafficking machinery involves additional elements. PCBP1/2 are iron carriers that bind iron in a 1:3 ratio and participate in intracellular iron trafficking. How PCBP1 acquires iron has not been elucidated. PCBP2 binds iron-loaded DMT1 and delivers iron directly to FPN1, facilitating thus both iron influx and efflux [51]. PCBP2 also carries iron released from heme by the action of HO-1 at the endoplasmic reticulum. Moreover, the labile iron pool may be regulated by PCBP1/2. PCBP1 and less so PCBP2 transport iron to ferritin [14]. PCBP1 also supplies iron to prolyl-hydroxylase (PHD2), the enzyme that initiates degradation of hypoxia-inducible factors (HIFs), one of the links between iron and hypoxia (see below) [52]. NCOA4 guides ferritin to the lysosomes for degradation. The liberated iron is transported to mitochondria by a not well-clarified mechanism mediated by DMT1, MCOLN1, and mitoferrins. TfR2 is also implicated in iron transport from the lysosome to mitochondria through a mitochondria-targeting motif present in the intracellular domain of TfR2. Due to this motif of TfR2, the lysosome approaches and physically interacts with a mitochondrion to facilitate iron transfer mediated by mucolipin [44,53].

Several elements of the uptake machinery are controlled by microRNAs. Thus, the miR-Let-7d and miR-16 decrease DMT1 expression. miR-485-3p, miR20a and miR-20b modulate the expression of FPN1. miR-200b induces decrease of ferritin, and miR-320 represses the expression of TfR1 [54].

Figure 2 presents the main aspects of cellular iron uptake and trafficking.

**Figure 2 biomedicines-13-00683-f002:**
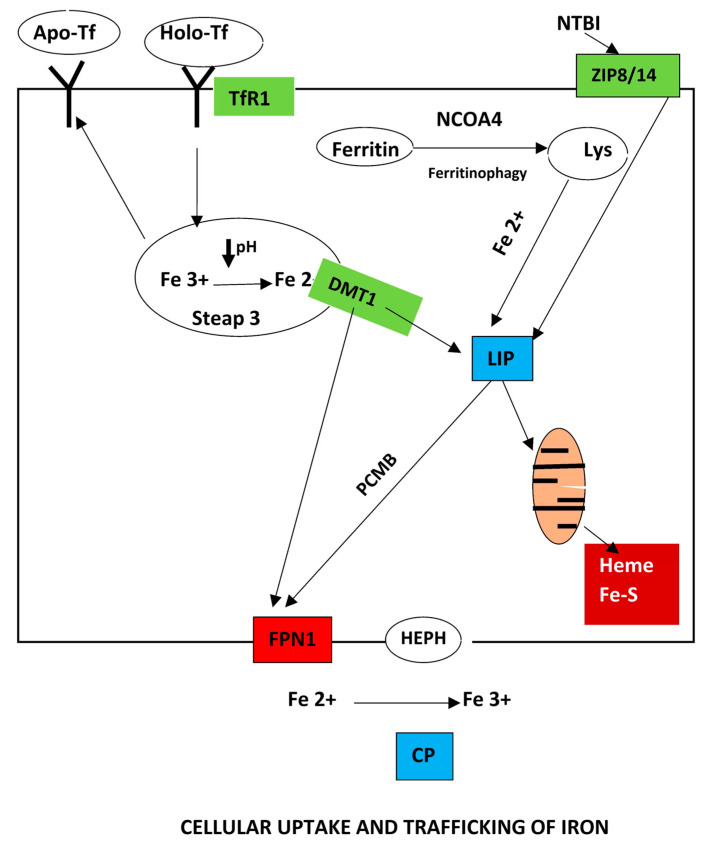
Cellular iron metabolism. Iron is taken up by cells either as heme (see Figure 1) or as non-heme iron. Holo-Tf binds to the TfR1 and the complex is transported into endosomes where the acidic pH leads to dissociation of the ferric iron from the complex. Fe^3+^ is reduced in its ferrous form by Steap3 and transported to the cytoplasm by DMT1. Iron from the labile iron pool is transported to mitochondria for heme and Fe–S cluster biosynthesis, and the remaining is either stored in ferritin or exported via ferroportin. The Tf/TfR 1 complex is recycled. The NTBI iron is taken up by the recently studied zinc transporters ZIP8 and ZIP14 and transferred to the LIP.

Cells with high iron turn-over have some additional interesting characteristics in iron handling.

#### 2.3.2. Macrophages

Adult humans have about 25 trillion RBCs and each second 2–5 million RBCs are recycled by a process called erythrophagocytosis in macrophages. This process covers almost 95% of the daily requirements of iron. Kupffer cells and splenic macrophages are the cells responsible for erythrophagocytosis [55,56]. Approximately 1.2 × 10^9^ heme molecules are found within a single RBC and the number becomes fascinating by the fact that RBCs are the most abundant cell type constituting almost 84% of human cells [57].

RBCs have a half-life of approximately 120 days. “Eat me” and “don’t eat me” ligands are implicated in the clearance of aging RBCs by macrophages. Opsonized phosphatidylserine (PS) is an important “eat me” ligand accumulated on the membrane of aging RBCs. It directly binds Stabilin-2, Tim-1, Tim-4 or CD300 on macrophages initiating a pro-phagocytic signal. Moreover, bridging molecules such as GAS-6 and lactadherin enhance RBC-macrophage phagocytosis by binding PS on RBCs and TAM receptors or αvβ3/β5 integrins on the macrophage. CD47–SIRPα interactions are a negative counteracting association that inhibit phagocytosis of RBCs by the macrophage [58,59].

After engulfment of RBCs, phagolysosomes are formed where degradation of RBCs takes place. Globins are hydrolyzed into amino acids and heme is released and transported to the cytosol by HRG1 and subsequently catabolized as mentioned before. Non-heme iron that is present in RBCs is transported by NRAMP1. Iron release is controlled by hepcidin. Erythrophagocytosis can be increased by inflammation and bacterial components such as LPS [60].

In addition to erythrophagocytosis, macrophages import iron from other sources such as Transferrin-bound iron, NTBI, hemopexin-bound heme and the haptoglobin–hemoglobin complex as mentioned above [61]. The lipocalin 2 (Lcn2) receptor 24p3R is present in macrophages. Siderophore iron that is associated with Lcn2 can be internalized and used by mitochondria to synthesize sulfur clusters and heme groups as previously described [62,63].

Ferroportin is regulated in macrophages in analogy with other cells. FPN1 transcription is inhibited by Bach1 and stimulated by Nrf2. Bach1 and Nrf2 bind MARE (Maf recognition elements) and ARE (Antioxidant Response Elements), respectively, in the *FPN* promoter. Erythrophagocytosis increases intracellular heme that enhances FPN1 transcription by ameliorating the inhibitory effect of Bach1 and by promoting the dissociation of Nrf2 from Keap1 as mentioned above. FPN1 mRNA is regulated by the IRPs (see below) and the miR-485-3p and miR-20a microRNAs. MiR-485-3p is initiated by iron deficiency [1,64,65].

It was recently demonstrated that TLR signaling selectively initiates the production of SpiC in tissue macrophages by an NF-κB-dependent mechanism. SpiC decreases the production of inflammatory cytokines and increases iron efflux by upregulating ferroportin in activated macrophages. This is reversed by increased levels of interferon-gamma during ongoing infection. The transcription factor Bach1 constitutively downregulates SpiC expression in macrophages [66].

Another important aspect on iron metabolism by macrophages is the result on their polarization. Modulation of iron regulation is influenced by the M1/M2 polarization [67].

The M1 pro-inflammatory macrophages have an iron storage phenotype, upregulating *HAMP* and genes encoding ferritin, while FPN1 and iron regulatory proteins are downregulated. By contrast, the IL-4–activated M2 macrophages upregulate CD163 and FPN1, while ferritin expression is downregulated [68,69,70]. The results of iron excess in macrophage polarization are controversial. Iron overload has been reported to favor M2 over M1 polarization followed by reduction in LPS-induced NF-kB nuclear translocation and a subsequent decrease of pro-inflammatory cytokines such as IL-1β and IL-6 [71,72]. Opposite results have been reported with overexpression of M1 markers and reduction of M2 markers in bone marrow macrophages and Kupffer cells [73].

HO-1 has also been involved as a promoter of M2 macrophage polarization [74,75]. Interestingly, there is an association between hepatocytes and macrophage polarization. Downregulation of FPN1 in hepatocytes induced polarization to the M2 phenotype increasing the production of IL-10 and TGF-β and consequently liver fibrosis [76].

Erythropoietin (EPO) also modulates macrophage polarization promoting the M2 phenotype via the EPOR/Jak2/STAT3/STAT6 signaling pathway in the presence of IL-4 [77].

Figure 3 presents the main aspects of iron regulation by macrophages.

#### 2.3.3. LSECs

LSECs take up iron through transferrin and NTBI. Internalization of iron increases oxidative stress and induction of BMP6 via Nrf2 [78,79]. The NTBI transporter, ZIP14 which is mandatory for NTBI uptake by hepatocytes, is not necessary for NTBI absorption by LSECs and Kupffer cells [80]. By contrast, the ZIP8 transporter has been identified as the main transporter for NTBI uptake by LSECs [81].

#### 2.3.4. Renal Interstitial Cells and Erythroferrone

Fundamental evidence demonstrated the role of erythropoietin (EPO) in hepcidin regulation [82], while the discovery of erythroferrone completed the mechanism of hepcidin regulation during hypoxia [83].

During hypoxemia, renal interstitial fibroblasts produce erythropoietin (EPO), a cytokine that promotes erythropoiesis in the bone marrow. EPO induces erythroferrone (ERFE) in erythroblasts, which is an erythropoietic inhibitor of hepcidin synthesis. The expression of EPO is transcriptionally initiated by HIF2α, which in turn is regulated by iron and oxygen through the activity of prolyl-hydroxylases (see below) [84,85,86]. ERFE inhibits hepatocyte BMP/SMAD signaling and hepcidin synthesis by impairing a BMP subgroup of BMP5, BMP6, and BMP7 [87,88]. Erythropoietin represses hepcidin in Bmp5se-mutant mice, but not in double Bmp5- and Bmp6-mutants [89]. The pathophysiology of ERFE is similar in rodents and humans [90].

Matriptase-2, which is stabilized in hepatocyte membranes in iron deficiency [91], overrules the effect of ERFE. ERFE cannot reduce the high hepcidin levels in MT2 knockout mice [92] or in MT2/TfR2 double knockout mice [93]. Erythroferrone and MT2 independently modulate hepcidin expression [94].

Very recent evidence indicated that the hepatokine fibrinogen-like 1 (FGL1) has a similar action with erythroferrone in hepcidin regulation. FGL1 is secreted by hepatocytes. FGL1 is an inhibitor of hepcidin that is produced in response to hypoxia in mice. FGL1 directly binds to BMP6 suppressing the BMP-SMAD signaling pathway leading to repression of *HAMP* [95,96]. Its role in vivo in humans remains to be established.

In addition to hypoxia, ER stress and steatosis upregulate FGL1 expression through the p38-C/EBP and STAT3 pathways. During acute inflammation, IL-6 induces FGL1. The binding of FGL1 to its cognate receptor LAG3 on T-cells prevents their activation [97].

### 2.4. Release

Intracellular iron levels are reduced by export via ferroportin, and by export of heme via feline leukemia virus subgroup C cellular receptor 1a (FLVCR1a). Interestingly iron is also exported by exosomes bound to ferritin [98,99,100].

The export of iron requires the presence of the ferroxidase hephaestin (Heph) which oxidizes the ferrous to ferric iron [40]. Soluble ceruloplasmin (CP) also mediates the oxidation of iron facilitating the binding to the plasma iron transporter transferrin (Tf). CP activity is important in humans, since mutations that reduce CP production (as found in aceruloplasminemia or copper depletion), lead to iron accumulation in the brain, liver and pancreas indicating an inverse relationship between copper and iron in the liver. Deficiency of iron leads to the loading of copper by a still-unknown mechanism. However, copper depletion causes hepatic iron loading in the liver due to decreased CP activity, which suppresses iron release [101].

The small amount (3–4 mg) of iron that circulates in plasma, bound to Tf, has a turnover rate of ~10 times/day to meet the requirements of iron for erythropoiesis [102,103].

Most importantly, ferritin iron can be used on demand through an autophagic process known as ferritinophagy [104]. The nuclear receptor coactivator 4 (NCOA4), a specific cargo receptor, is implicated (Fig2). Ferritin is transported to lysosomes for degradation and iron is released upon binding to NCOA4 as mentioned above [105]. Ferritinophagy is critical for erythropoiesis as it mobilizes iron from macrophages and developing erythroid cells [106,107]. Intestinal ferritinophagy through NCOA4 is a target for HIF2a (see below) leading to the integration of the NCOA4-HIF2a axis in the systemic iron needs [108].

### 2.5. Major Regulators of Iron Metabolism

The three elements of this system are the central mechanism of iron regulation.

#### 2.5.1. BMPs

Bone morphogenetic proteins (BMPs) are growth factors belonging to the transforming growth factor (TGF)-β superfamily, which also includes TGF-βs and activins. The most extensively studied are BMP2 and BMP6 [109,110]. They mostly originate from liver sinusoidal endothelial cells (LSECs) and their deletion leads to iron overload in mice due to reduction of hepcidin mRNA expression. BMP2 has a critical role in the regulation of hepcidin production. Selective suppression of BMP2 in LSECs seriously impairs hepcidin expression through downregulation of the BMP-SMAD axis leading to body iron overload. On the other hand, selective deletion of BMP6 in Kupffer cells or in hepatocytes did not significantly change hepcidin expression or tissue iron overload [111,112]. LSECs also release the BMP-binding Endothelial Regulator (BMPER) that interferes with the BMPs signaling pathway, leading to hepcidin downregulation in liver fibrosis [113]. Iron is captured by unknown receptor(s) on the surface of LSECs leading to increased mitochondrial reactive oxygen species (ROS), which activates Nrf2 to upregulate BMP6 transcription [114].

The receptors implicated in the BMPs pathway are the activin receptor-like kinases (ALKs) which are serine-threonine kinases. They are classified into type I (BMPRI) and type II (BMPRII) receptors. BMPRIIs are constitutively active while BMPRIs need activation by BMPRIIs after phosphorylation of their intracellular glycine/serine-rich domain. There are four BMPRIs (ALK1, ALK2, ALK3 and ALK6) and three type-II receptors (ACVR2A, ACVR2B and BMPR2). The two receptor types act as dimers. The formation of a hexameric complex, made of two BMPRIs, two BMPRIIs and a dimer of ligands, is required to activate the BMP-SMAD pathway [115].

Secreted BMP6 and BMP2 act together, most likely as a heterodimer (BMP2/6), to bind to BMP receptor complexes on the hepatocyte membrane containing two type I receptors (ALK3 homodimers or ALK2/3 heterodimers), two type II receptors (ACVR2A and/or BMPR2) and co-receptor hemojuvelin (HJV) [116]. This complex induces signals leading to the phosphorylation of SMAD proteins. The BMP-mothers against the decapentaplegic homolog 1/5/8 (SMAD1/5/8) pathway is the critical regulatory mechanism of hepcidin production. The phosphorylated complex after recruitment of SMAD4 is translocated to the nucleus and binds to the BMP-responsive elements in the hepcidin (*HAMP*) gene promoter. Upon the binding of iron-saturated holo-Tf, the transferrin receptor 2 (TfR2) is stabilized and enhances hepcidin upregulation by interacting with the BMP/SMAD pathway [44,117].

It has been suggested that ALK2 is mainly implicated in BMP6-dependent hepcidin synthesis in conditions of iron excess and is inhibited by the immunophilin FKBP12 [118], whereas ALK3 is involved in basal hepcidin expression and mostly signals in response to BMP2 [84].

HJV and TfR2 are required for the canonical BMP-SMAD pathway. However, HH proteins are not required for the activation of BMP-SMAD signaling and hepcidin expression by BMP2, particularly in the case of an acute increase of BMP2-induced hepcidin expression in response to increased iron levels [119]. It has been recently demonstrated that both ALK2 and ALK3 are prerequisites in vivo for the HJV-mediated initiation of hepcidin production [120]. This is in accordance with a previous report where combined hepatocyte deletion of ALK2 and ALK3 aggravates iron overloading than deletion of ALK3 alone [121]. BMP2 and BMP6 may act in concert to regulate hepcidin expression, but BMP2-independent and BMP6-independent SMAD1/5/8 phosphorylation may also contribute to hepcidin regulation by iron. Importantly, HFE partly regulates hepcidin through a BMP2-independent but SMAD1/5/8-dependent mechanism [122].

Other factors may also be implicated in the BMP/SMAd pathway. Thus, c-Jun participates in the BMP6 regulation independent of Nrf-2, as deletion of the c-jun encoding gene in LECs blocks BMP6 but not Nrf2. The c-jun signaling may be a redundant pathway [123]. Additional BMPs may also participate to hepcidin regulation. BMP4 also has a high affinity to HJV and initiates hepcidin expression in hepatocyte cultures. The BMP4 p.H251Y and p.R269Q variants repress hepcidin production by inhibiting the BMP/SMAD axis, indicating a possible role in iron overload [124].

A different mechanism of hepatic BMP6 gene expression has also been reported. The orphan nuclear receptor, estrogen-related receptor γ (ERRγ), was increased after IL-6 stimulation leading to upregulated BMP6 expression. Overexpression of ERR alone was able to upregulate BMP6 indicating that IL-6 acts upstream of ERRγ [125].

The E3 ubiquitin-protein ligase SMURF1 was recently found to specifically regulate the BMP/SMAD pathway by upregulating the responsiveness of hepatocytes to BMPs during iron excess [126].

Finally, the important regulation of BMP6 by Nfr2 should be stressed. Nrf2 activation stimulates the BMP6–hepcidin axis in iron-loaded mice providing a connection between protection from oxidative stress and iron overload. Activation of Nrf2 inhibits iron loading in hemochromatosis and thalassemia [114].

The so-called ‘toxic iron’ and not iron itself is the factor initiating BMP6-driven hepcidin synthesis and a distinction should be made between iron accumulation that is tolerated, and the development of pathogenic iron overload [1]. Moreover, recent findings suggest that iron may also be able to directly and negatively affect LSEC phenotype. Nevertheless, a persistent activation of Nrf2 has been reported to inhibit autophagy [127] and, since autophagy has a protective role in maintaining LSECs integrity and phenotype (i.e., fenestrae), it is conceivable that iron-induced Nrf2 activity may contribute to endothelial dysfunction [127,128].

#### 2.5.2. Hepcidin

The association of hepcidin with iron overload was initially described almost 25 years ago [129]. Later on, the regulation of hepcidin by BMPs was established [130,131].

Hepcidin is initially generated as a pre-pro-peptide of 84 amino acids, from which the 60 amino acid pro-hepcidin, and then the 25 amino acid active forms are produced [132].

The liver is the central regulator of iron homeostasis being the producer of hepcidin which is the most important factor in iron turnover. Hepcidin acts on both the iron export protein FPN1 and the iron importer DMT1. It is well established that hepcidin secretion by the hepatocyte leads to internalization and degradation of FNP1 by lysosomes. In cells with a high iron turnover such as enterocytes, macrophages and hepatocytes, reduction of FPN1 expression leads to intracellular iron sequestration and reduction of circulating iron [133,134]. Additionally, hepcidin decreases the expression of DMT1 on the apex of enterocytes, and thus decreases intestinal iron absorption [135].

BMP production and secretion is the first step in hepcidin biosynthesis as detailed before [136]. The second step in hepcidin synthesis takes place in the hepatocyte. Holo-Tf binds to the transferrin receptors TfR1 and TfR2 on the surface of hepatocytes. The association of TfRs with HFE for hepcidin regulation depends on the competitive interaction between HFE and TfR1 or TfR2. Transferrin iron interacts with high affinity to TfR1, compared with the affinity to TfR2. Therefore, under low transferrin saturation, transferrin iron and HFE bind only to TfR1, and hepcidin expression is not initiated. TfR2 is degraded by lysosomes [43]. Transferrin iron interacts with TfR2 when Tf iron is increased leading to dissociation of HFE from TfR1 and attachment to TfR2 to initiate hepcidin transcription [137]. The complex binds to the BMP-responsive element on *HAMP* activating the transcription of hepcidin [138]. This is in accordance with recent findings in a murine model where inappropriately high hepcidin levels were observed in hepatocytes of Tfr1 deficient mice. This was attributed to increased activation of HFE signaling for hepcidin production due to the absence of TfR1 function to segregate HFE [139]. The HFE–TfR2 complex seems to be a sensitive hepatic iron sensor [43].

Neogenin is required as a stabilizer of HJV and the BMP receptor complex. Neogenin binds to a distinct domain from BMP ligands and functions as a scaffold protein to assist the complex formation. Neogenin mutant mice have iron overload due to impaired BMP signaling and hepcidin repression [140,141]. Moreover, neogenin suppresses the BMP-2-induced phosphorylation of the Smad1/5/8 complex [142] and favors the cleavage of HJV by matriptase-2 or furin [54,143]. The hepatocyte Smad1/5/8 knockout murine model of hemochromatosis revealed an important role of SMAD8 in hepcidin regulation and demonstrated that testosterone and epidermal growth factor (EGF) require SMAD1/5/8 to regulate hepcidin production [144].

Similar murine models established that SMAD2 promotes the phosphorylation of SMAD 1/5/8 while SMAD6/7, endofin and ATOH8 reduce the function of the BMP/SMAD axis [145]. Thus, genetic deletion of SMAD7 in hepatocytes leads to mild iron-restricted anemia with reduced iron stores due to hepcidin upregulation [146]. On the contrary, SMAD7 upregulation in hepatocytes causes iron overload as a consequence of hepcidin inhibition [145].

HFE is an important protein implicated in hepcidin biosynthesis and HFE mutations have been associated with the majority of hereditary hemochromatosis cases. The detailed molecular mechanisms HFE hemochromatosis are not fully delineated. HFE is not implicated in the regulation of BMP6 by iron. HFE regulates hepcidin production in response to iron by interfering with the BMP-SMAD1/5/8 pathway downstream of BMP6 as liver BMP6 mRNA was normally induced by iron, but liver SMAD1/5/8 signaling was abnormally suppressed relative to BMP6 mRNA in HFE knockout mice [147,148]. HFE is also initiated by iron overload, suggesting that it has an iron sensing function. HFE predominantly uses the BMP type I receptor ALK3 to initiate hepcidin expression [149,150]. Interestingly, an ablation of the HFE localizing protein beta-2-microglobulin (B2M) promotes iron overload by reducing liver SMAD1/5/8 signaling in BMP6 KO mice. Hepcidin deficiency in this model indicates that a BMP6-independent mechanism is involved in the HFE intersection with the SMAD pathway to regulate hepcidin [151].

An important element in hepcidin secretion is the transmembrane serine protease matriptase 2 (MT-2, encoded by *TMPRSS6*) that cleaves HJV from the membrane surface to suppress hepcidin production and upregulate FPN1 activity [152]. MT2 is synthesized as a zymogen and is activated by autocleavage.

Mutations of MT2 unable to cleave proteins can still inhibit hepcidin expression. The extracellular domain of TMPRSS6 is essential for the suppression of hepcidin expression. It has been suggested that the ability of TMPRSS6 to bind other proteins, but not cleave other proteins, is a critical determinant of its function [153,154].

However, MT2 effects are more complex than a simple cleavage of HJV. Indeed, it was reported that MT2 represses hepcidin expression independently of HJV as it cleaves ALK2, ALK3, ActRIIA, BMP2, HFE, TFR2 and, to a lesser degree HJV. Therefore, MT2 inhibits hepcidin by cleaving several components of the hepcidin initiation pathway [155].

In summary, recent evidence indicate that repression of BMP expression, HJV cleavage from the membrane, sequestration of HFE with TfR1 and reduced TfR2 expression all impair the BMP-SMAD pathway leading to reduced hepcidin expression [40,102,117,156].

Other factors regulating hepcidin. It is conceivable that a critical regulator of iron metabolism such as hepcidin would be influenced by a large variety of other factors both hereditary and acquired.

The immunophilin FKBP12 is an additional hepcidin regulator. After binding to the receptor ALK2, it represses the BMP/SMADs pathway [118,157].

Hepcidin is also downregulated by the production of soluble factors such as growth differentiation factor 15 (GDF-15) and twisted gastrulation protein homolog 1 (TWSG1) by proliferating erythroblasts. GDF-15 and TWSG1 inhibit BMP6 or induce MT2 [158].

Recent evidence demonstrated a role for the mechanoreceptor PIEZO1 in the regulation of iron metabolism. Overexpression in hepatoma cell lines of the R2456H and R2488Q gain of function (GoF) PIEZO1 mutants led to inhibition of the BMP/SMADs pathway, and decreased expression of the *HAMP* gene.

Interestingly, individuals of African descent with increased plasma iron were harboring the E756del GoF PIEZO1 allele in their macrophages [159].

Hepcidin production is altered during inflammation through the action of interleukins [160].

The main inflammatory cytokine affecting the hepcidin pathway is IL-6 which is produced by Kupffer cells and other macrophages. IL-6 binds to receptors on hepatocytes promoting their dimerization and activation of JAK1/2, which phosphorylate STAT3. Eventually, phospho-STAT3 translocates to the nucleus and binds to the signal transducer and activator of transcription (STAT)-BS on hepcidin promoter [161,162]. The other inflammatory cytokine that upregulates hepcidin is IL-1β. Induction of CCAAT-enhancer binding protein (C/EBP) δ and IL-6 in response to IL-1β treatment stimulates hepcidin transcription via the C/EBP-binding site (C/EBP-BS) on the hepcidin promoter. Interestingly, lipopolysaccharide (LPS) initiates IL-1β by Kupffer cells in the murine liver and in cultured macrophage cell lines [163].

IL-1β induces hepcidin production by a second pathway. IL-1β phosphorylates c-Jun N-terminal kinase (JNK) and its substrate JunB leading to hepcidin transcription through binding of JunB to the cAMP response element (CRE) site B, on the hepcidin promoter [164]. Several other cytokines, such as IL-22 [165], oncostatin M [166] and leukemia inhibitory factor (LIF) [166], upregulate hepcidin production, but their precise role in iron regulation in humans has not been clarified.

A recent interesting study addressed the hierarchy of iron and mild inflammation in the regulation of hepcidin in humans. Iron-deficiency anemia is dominant over inflammation. Moreover, in the absence of anemia, inflammation increased serum hepcidin but had no effect on enterocyte iron absorption, indicating that macrophages are more sensitive compared to enterocytes to increased inflammation-associated hepcidin [167]. This contrasts with previous mice findings where the inflammatory pathway overrides iron deficiency [168].

Other signals that modulate hepcidin include ER stress initiated by protein misfolding. ER stress upregulates hepcidin transcription through cyclic AMP-responsive element-binding protein 3-like protein 3 (CREB3L3), or through hormones and growth factors [1].

Glycochenodeoxycholate (GCDCA) is also an upregulator for hepcidin, possibly acting through stimulation of the farnesoid X receptor (FXR) and the BMP6/ALK3-SMAD pathway [169].

As mentioned before, testosterone represses hepcidin production through activation of the epidermal growth factor receptor (EGFR) leading to inhibition of the SMAD1/5/8 signaling pathway [170]. On the contrary, progesterone induces hepcidin production independent of the BMP/SMAD pathway [171]. Estrogen data are controversial. The fibroblast growth factor 23 (FGF23), which is a regulator of phosphate and mineral metabolism, was reported to inhibit BMP6 or IL-6-moderated hepcidin production [172].

Hepatitis C virus (HCV) also modulates hepcidin production. Two proteins of the virion are antagonistic. HCV core protein is a hepcidin inducer, while NS5A reduces hepcidin levels [173,174].

The increased iron load in cells caused by HCV infection was also attributed to a similar mechanism with ER stress. HCV infection activated the CREB3L3, which not only acts on the hepcidin promoter but also increases BMP6 expression. In addition, the NS3-4A serine protease of the HCV cleaves FPN1 and reduces iron export from the cells [175].

Other liver diseases also impair hepcidin production. Cholestasis represses hepcidin levels by inhibiting the IL-6/STAT3 pathway while the decreased level of hepcidin in autoimmune liver disease has not been explained [176].

Alcohol inhibits hepcidin production either directly acting on C/EBP or indirectly through the toll-like receptor 4 (TLR4) effect on non-parenchymal liver cells. The signal responsible for this effect has not been determined. Nonetheless, in a murine model of defective TLR4 receptors, alcohol does not repress hepcidin expression [177]. Kupffer cells and hepatocytes are possibly not implicated in alcohol-induced hepcidin modulation [178].

##### Posttranscriptional Hepcidin Modification

The BMP/SMAD pathway can be posttranscriptionally modulated by micro RNAs (miRNAs) which bind to the 3′ untranslated region (UTR) of genes and inhibit their translation. Two miRNAs are interfering with hepcidin expression. Depletion of miR-122 in mice resulted in systemic iron deficiency. Moreover, miR-122 inhibition increased mRNA from genes that regulate iron turn over, such as *HFE*, *HJV* and *HAMP*. MiR-122 directly targeted the 3′ UTR mRNAs that encode activators of hepcidin such as *HFE* and *HJV* [179]. On the other hand, miR-210 blocks the cleavage of HJV by MT2 and increases hepcidin production [54]. MiR130a is increased in iron deficiency in mice and contributes to hepcidin reduction by targeting the 3′ UTR of ALK2 [180].

Histone modifications also participate in the modulation of hepcidin expression. The activation of hepcidin promoter by SMAD4 is associated with a modification of histone H3 to a transcriptionally active form [181]. Moreover, repression of hepcidin expression is associated with epigenetic modulation by histone deacetylase 3 [182].

Figure 4 depicts hepcidin regulation by LSECs and hepatocytes.

#### 2.5.3. Ferroportin (FPN1)

Binding of hepcidin to FPN1 leads to FPN1 internalization, followed by lysosomal degradation in vitro as mentioned before [183]. Recent evidence indicates that hepcidin also occludes the central cavity of ferroportin to block iron export. Inhibition of FPN, by degradation or occlusion by hepcidin, reduces iron export into the circulation, reducing thus the amount of non-transferrin bound iron (NTBI) [184,185,186]. Ferroportin is mostly localized in duodenal enterocytes, Kupffer cells and splenic macrophages. However, FPN1 from these cells differ in the apparent molecular masses possibly due to different *N*-linked glycosylations. In addition, FPNs in enterocytes and in bone marrow-derived macrophages have a different membrane localization. These differences do not prevent a coordinate expression of FPN1 in both enterocytes and macrophages [187].

FPN1 expression may also be modulated independently of hepcidin effects. Infections with intracellular pathogens activate Nrf2 and upregulate ferroportin transcription. Several other transcription factors such as SpiC, MTF1 and HIF-2 activate ferroportin transcription, while unidentified factors inhibit FPN1 transcription during inflammation [64,188]. In murine models, stimulation of TLR2/TLR6 decreases FPN1 messenger RNA and protein expression in macrophages including Kupffer cells, without alterations of hepcidin expression. Moreover, FPN1 mutant mice with an interrupted hepcidin/ferroportin association reduce FPN1 expression after stimulation of TLR2/TLR6 indicating that FPN1 reduction and not hepcidin is the first response to iron restriction during pathogen invasions [189].

Posttranslational modulation of FPN1 expression has also been described. FPN1 intracellular trafficking depends on the small ubiquitin-like modifier (SUMO). Sumoylation-defective mutations lead to low intracellular iron content [190]. Moreover, FPN1 is involved in autophagy, and autophagy-dependent FPN1 degradation inhibits iron release promoting ferroptosis [191,192].

Interestingly, enterocyte FPN1 expression is inversely related to total body iron, but this is not the case in the liver, where FPN is low during low iron conditions and high during iron overload. This is possibly due to the fact that there are two FPN1 splice variants, one that incorporates 5′ iron responsive element (IRE) and one without IRE. High iron levels will prevent the iron regulatory protein (IRP) to bind IRE and translation will be intact. On the other hand, low iron will allow the binding leading to inhibition of FPN1 translation. During iron deficiency in enterocytes, translation from the IRE+ variant is reduced, but expression of the IRE-variant is increased promoting iron absorption [101].

There are two major regulators of iron metabolism that require a more detailed presentation. The iron regulatory proteins (IRPs) mediate the expression of several proteins involved in iron metabolism [193] and the hypoxia-sensing machinery operated by the hypoxia inducible transcription factors (HIFs) [194].

#### 2.5.4. Cellular Regulation of Iron by IRE/IRPs

The IRE/IRP system mediates the expression of DMT1, TfR1, FPN1, and ferritin. This system has been most extensively investigated in macrophages and hepatocytes [195]. The function of the iron-regulating proteins IRP1 and IRP2 is to control the translation of cytosolic proteins implicated in iron uptake, trafficking and export reacting to cytoplasmic iron levels. This is achieved by binding to iron-responsive elements (IREs) in the mRNA of target genes [196]. The 30-nucleotide long RNA motifs IREs are found in either the 3′ UTR or 5′ UTR of the target mRNA. When iron levels within the cell are adequate, IRP1 may function as aconitase transforming citrate to isocitrate to balance the amount of NADPH dehydrogenase with the amount of acetyl-CoA [195]. When iron levels are low, IRPs bind to IREs in the 3′ UTR of TfR1 mRNA inhibiting its degradation. On the other hand, binding of IRP1 in the 5′ UTR ferritin and ferroportin mRNAs as well as in the rate-limiting enzyme of heme biogenesis ALAS (aminolevulinic acid synthase) results in inhibition of their translation. The resultant increase in TfR1-mediated iron uptake and the decrease in iron storage and export enhances the labile iron pool [1,197]. When iron levels normalize again, IRP1 regains its Fe-S cluster and aconitase activity [197].

In iron overload, IRP2 associates with FBLX5 (F-Box and Leucine-Rich Repeat Protein 5), and an E3 ligase complex leading to IRP degradation [198]. FBXL5 is stable in iron-rich cells and degraded in iron-depleted cells. The iron-binding N-terminal hemerythrin-like (Hr) domain of FBXL5 functions as an iron sensor. The free-from-iron Hr domain increases FBXL5 degradation, resulting in IRP2 accumulation [199,200]. Intriguingly, some isoforms of the DMT1 and FPN mRNAs lack the IRE and thus escape IRP regulation [201]. The expression of an FPN mRNA splice variant lacking IRE allows enterocytes to export iron under conditions of cellular iron depletion [201,202], as mentioned before.

#### 2.5.5. Hypoxia

Iron metabolism is also modulated in response to hypoxia by HIF1α and HIF2α. If adequate amounts of ferrous iron and oxygen are present, HIF1α and HIF2α are hydroxylated by prolyl hydroxylases (PHDs) encoded by the three genes *EGLN1-3* and are then degraded by Ligase E3 [203,204]. In contrast, low amounts of intracellular iron levels or hypoxia induce stabilization and sequestration of HIF1α and HIF2α in the nucleus triggering the recruitment of HIF1β. The HIFα/HIFβ complex binds to hypoxia-responsive elements (HRE) increasing the transcription of genes implicated in iron metabolism such as *Tf*, *TfR*, *DMT1*, *FPN*, *DCYTB*, *ceruloplasmin* and *HO-1* [205]. HIF2α is reduced by bacterial metabolites of the microbiome, and negatively regulated by IRPs, mostly by IRP1 [206].

Hepcidin is also controlled by oxygen levels. HIF either directly inhibit *HAMP* promoter or indirectly by activating MT2 [54]. At the same time, hepcidin controls HIF-2α in iron depletion and iron overload. Hepcidin inhibition of enterocyte FPN1 also inhibits HIF-2α expression, by raising intracellular iron followed by activation of iron-dependent prolyl hydroxylases. In iron depletion, hepcidin reduction and increased efflux of iron decreases the activity of prolyl hydroxylases leading to stabilization of HIF-2α, and activation of DcytB, DMT1 and FPN1 [207,208].

An increased prevalence of the HIF1A p.Phe582Ser and p.Ala588Thr variants was reported in patients with severe iron overload. HIF1A was shown to be a modifier of the HFE-HH phenotype collaborating with HFE in the inhibition of hepcidin synthesis [209].

HIF-1 is not essential for macrophages erythrophagocytosis [210]. HIF-2 on the other hand is critical for adult erythropoiesis, by regulating EPO but also through iron mobilization. In the liver, HIF-2 activation inhibits hepcidin production through an EPO-mediated stimulation of erythropoiesis [211]. HIF-2 also participates in iron deposition of hemochromatosis [212].

In conclusion, the IRP/IRE and HIF systems are parts of a multilayered regulation of iron metabolism. Regulatory components either collaborate or interact. FPN1 regulation is possibly the best example. It is modulated at the transcriptional (via Bach1/Nrf2 or HIF2), posttranscriptional (via miR-485-3p), translational (via the IRPs), and posttranslational (via hepcidin) levels. HIF2α and IRPs have similar targets (FPN1 and DMT1), while HIF2α is modulated by the IRPs. Several details of the regulatory mechanism have not been clarified so far [213,214]. Extensive reviews of iron metabolism have recently been published [99,206,215].

## 3. Hemochromatosis

Hereditary hemochromatosis (HH) is a genetic iron disorder characterized by intestinal iron hyper absorption with excessive iron deposition in several organs with serious pathological consequences such as liver cirrhosis, diabetes mellitus, cardiomyopathy, arthropathy and skin pigmentation [162,216]. Hepatocytes and pancreatic acinar cells internalize NTBI via the transporter ZIP14 and become iron overloaded [80]. HFE-linked hemochromatosis patients have a limited iron storage in enterocytes, Kupffer cells and macrophages of spleen and bone marrow [217,218].

Hemochromatosis was first reported in 1865 by the French physician Armand Trousseau as “bronze diabetes” [219] due to the skin pigmentation of some diabetic patients. Association with iron overload was conducted in 1890 by the German pathologist Friedrich Daniel von Recklinghausen, who first used the term “hemochromatosis” (from the Greek words aíμa = blood and χρώμa = color) and proposed that iron overload may damage the endocrine functions of the pancreas [220]. The hereditary nature of hemochromatosis was demonstrated by the group of Marcel Simon and colleagues in the late 1970s [221,222].

Genetics were described in the original paper by Feder et al. [223]. The majority of patients were homozygous for a substitution of tyrosine for cysteine at position 282 (C282Y) of the HFE polypeptide. This mutation does not allow for correct folding of HFE by impairing the association with β2 microglobulin (B2M) and its expression on the cell surface. HFE was shown to be a membrane protein of the major histocompatibility complex class I family that hetero-dimerizes with β2-microglobulin [224] playing an immunological role in impairing MHC-I antigen presentation and T cell activation [225]. The C282Y HFE mutation provides advantage to infection with intracellular pathogens due to increased iron load [226,227].

The 343 amino acid HFE protein has many variants. The three most extensively studied are H63D, C282Y and S56C. HH is the most common genetic disorder of Europeans being almost exclusively a Caucasian disease of northwestern European origin. The highest frequency of homozygosity for this condition is found in Ireland (1:83), but is prevalent (approximately 1:200) in other parts of Europe and worldwide where there are people of northern European ancestry such as Australia [228,229]. HH type 1 is mostly related to two *HFE* gene mutations. Approximately 95% of affected individuals have the p.C282Y (p.Cyst282Tyr) mutation and 4% have the p.C282Y/p.Hist63Asp compound heterozygote genotype [229]. The distribution of C282Y mutation is consistent with the theory of a Celtic origin of the mutation. The highest percentages are found in the UK and Brittany with a decreasing gradient from the north to south of Europe with the lowest percentage found in Italy (64%) and Greece (50%) [230,231]. In the USA, the prevalence of heterozygosity (C282Y/no p.C282Y mutations) was 9.5% in non-Hispanic white people, with a lower prevalence in non-Hispanic black people (2.3%), and Mexican-Americans (2.8%) while 0.3% of studied non-Hispanic white people were p. C282Y homozygotes according to the third National Health and Nutrition Examination Survey (NHANES). The H63D variant was present in 15–20% of Caucasians [232].

An additional study of almost 100,000 adults over a period of 5 years from the USA and Canada (HEIRS study) estimated that the prevalence of p.Cys282Tyr homozygosity was 0.44% in white non-Hispanic people, 0.11% in native Americans, 0.027% in Hispanic, 0.014% in Blacks, 0.012% in Pacific Island descendants and 0.000039% in those with Asiatic ancestry [233]. In a selection of 27 studies, 6302 subjects of European countries showed an average prevalence of 0.4% for p.Cys282Tyr homozygosity and 9.2% for p.Cys282Tyr heterozygotes [234]. A more recent cohort study of 451243 individuals of European descent from 22 centers in England, Scotland and Wales showed 0.6% of p.Cys282Tyr homozygosity. Overt haemochromatosis was diagnosed in 21.7% of men and 9.8% of women of the homogygous individuals over a mean follow-up of seven years [235].

HFE-related HH presents with a variable phenotype. Body iron varies from normal values up to serious iron overload associated with organ involvement. The majority of predisposed individuals usually show increased transferrin saturation and serum ferritin, but only a minority have clinical symptoms and severe tissue damage. HFE-related HH has therefore, a high biochemical penetrance, but a low clinical prevalence which may be related to other comorbidities [236]. A review of hemochromatosis penetrance in the USA found that for one million C282Y homozygotes, up to 38–50% will have biochemical signs of iron overload and 10–33% will develop hemochromatosis-associated morbidity [237]. Other studies reported a similar clinical penetrance of less than 30% in males and approximately 1% in females indicating that C282Y is not pathogenic per se. It is possibly a genetic variant that favors iron overload in association with gender, alcohol consumption and other genetic variants [238,239].

Several genes such as *HAMP*, *BMP2*, *BMP4*, *HJV*, *BMP6*, *TMPRSS6*, *Ceruloplasmin* and *Tf* that may modify the penetrance of HFE-HH have been identified in murine models and patients [162].

Current evidence showed that hereditary hemochromatosis incorporates five different genetic forms with a common characteristic of the high transferrin saturation and serum ferritin but with different clinical penetrance. Four of these forms involve other genes and not the *HFE* gene. They are non-HFE hemochromatosis type 2A (*HFE2*, encoding HJV), type 2B (*HAMP*, encoding hepcidin), type 3 (*TfR2*, encoding transferring receptor-2) and types 4A and B (*SLC40A1,* encoding ferroportin). Loss-of-function mutations of *SLC40A1* cause HH type 4A (ferroportin disease), while gain-of-function mutations of *SLC40A1* cause HH type 4B leading to hepcidin resistance. Pathogenic alleles are very rare, estimated to be 74/100,000 for type 2A, 20/100,000 for type 2B, 30/100,000 for type 3 and 90/100,000 for type 4 non-HFE hemochromatosis [240,241,242].

HH types are also classified according to age of onset as type 1 (HFE variants, adulthood onset), type 2 (*HJV* or *HAMP*, juvenile, the most severe clinically), type 3 (*TfR2*, before the age of 30 years) and type 4 (FPN gain-of-function variants, adulthood) [216,243,244]. Only a fraction of patients with HH type 1 develop iron overload. This is due to the weak effect of HFE in the induction of hepcidin expression. Moreover, *HFE* mutations are not significantly associated with the severity of liver fibrosis [245].

Additional iron-loaded inherited diseases include aceruloplasminemia, which is caused by defective iron release. Iron retention in the brain causes severe neurological symptoms. Atransferrinemia and DMT1 deficiency are other diseases characterized by iron deficient erythropoiesis, and parenchymal iron overload due to secondary hepcidin suppression [246].

New variants associated with iron overload are constantly described. The p.Leu96Pro *BMP6* mutation, not related with the well-established five hemochromatosis genes, was identified in five unrelated families in France. Several carriers of this mutation had mild to moderate iron deposition. Severe iron overload was demonstrated in two patients with additional comorbidities such as obesity, metabolic syndrome and the carriage of the *HFE* p.His63Asp mutation in a woman [247]. *BMP6* heterozygous mutations were also identified in four individuals from three Italian families, where two new mutations (p.Glu112Gln, p.Arg257His) were described in addition to the p.Leu96Pro. Comorbidities such as alcoholism, the metabolic syndrome and β-thalassemia minor (one patient) were also present. Unfortunately, in both studies, no liver biopsy was available to assess the iron status of Kupffer cells [248].

A recent meta-analysis of three genome-wide association studies from Iceland, the UK and Denmark revealed 46 new loci associated with iron metabolism awaiting elucidation of their actual function. Interestingly, the common missense variant in *TMPRSS6* (rs855791A) protects against HH (OR = 0.80), being a new modifier of this disorder [249]. Due to this constant identification of new variants related to HH, a new classification was recently proposed (Table 1).

### 3.1. Current Pathogenesis of Hemochromatosis

The underlying pathophysiologic impairment that is common in all forms of HH is hepcidin deficiency accompanied by FPN1 over activity due to mutations in the genes encoding hepcidin itself *(HAMP*) FPN1, or other elements of hepcidin regulation by iron such as HFE, TfR2 or HJV [9,134].

The specific molecular mechanisms leading to HH by HFE mutations have not been clarified. Initially, two theories were proposed to explain how an abnormal HFE protein causes iron overload. They are not mutually exclusive. The hepcidin hypothesis proposed that the abnormal HFE was associated with TfR on the surface of hepatocytes resulting in low hepatocellular hepcidin production, and increased iron absorption by the enterocyte as hepcidin is a negative regulator of dietary iron uptake. The duodenal crypt cell programming hypothesis suggested that the HFE protein, associated with TfR on the surface of crypt enterocytes, facilitates absorption of iron into the crypt enterocyte, a process that an abnormal HFE protein fails to do. A relatively iron-deficient enterocyte tries to over-absorb iron to accommodate for the relative deficit [253,254,255]. Low hepcidin in HFE-HH stabilizes FPN1, leading to increased iron efflux from macrophages and duodenal enterocytes. Therefore, macrophages in HFE mutations are relatively iron-depleted despite systemic iron overload [133].

Today, the hepcidin deficiency theory has been further elaborated based mainly in the BMPs regulation of hepcidin production as detailed above. Type 1 HFE-HH impairs the assembly of the iron-sensing complex, as transferrin binding to TfR1 normally dissociates HFE to interact with TfR2. Mutations of the *HJV* (type 2A HH) and *TfR2* (type 3 HH) genes, also encode impaired elements of the iron-sensing complex. Mutations of the *HAMP* gene (type 2B HH) cause decreased hepcidin levels leading also to FPN1 upregulation [8,242]. This is followed by increased iron absorption and macrophage iron efflux that eventually exceeds the transporting capacity of transferrin, causing NTBI production and iron accumulation in parenchymal tissues. The differences among the forms of primary iron overload are quantitative rather than qualitative (amount of iron deposited, hepcidin levels) [241,246].

A functional abnormality of HFE due to ablation of beta-2-microglobulin in a BMP6 KO murine model further lowers liver SMAD1/5/8 signaling, increasing hepcidin reduction, and iron overload. It seems, therefore, that a downstream to the BMP6 mechanism is available for HFE to interact with the SMAD pathway and regulate hepcidin synthesis [151]. In HH with *HFE* gene mutations, hepcidin synthesis is inappropriately low relative to iron load. Hepcidin levels may be normal at the time of diagnosis, but not sufficient according to iron excess. As iron decreases to normal levels by venesections, frank hepcidin deficiency becomes evident [256,257]. By contrast, expression of BMP6 was appropriately elevated in HFE HH in relation to iron overload. However, decreased phosphorylation of Smad1/Smad5/Smad8 protein relative to iron load was found. Furthermore, Smad6 and Smad7, the inhibitors of BMP signaling, were upregulated in HFE-HH indicating that it is the signaling pathway downstream of BMP that underlies hepcidin relative deficiency in HFE-HH [258].

The pathophysiology of liver damage in HH is also not fully clarified and only data from animal models are available. In a murine model of HFE-HH, no liver damage was observed despite the high iron deposition [259,260]. Only when TfR2 deletion was added, liver fibrosis was developed. However, the responsible mechanisms were not examined.

Earlier studies demonstrated that liver injury is associated with increased lysosomal iron leading to lysosomal rupture and release of the metalloprotease cathepsin B into the cytoplasm that may cause hepatocyte damage [261,262]. Increased p62 levels indicated that increased autophagy was responsible for the lysosomal iron influx [263].

Liver fibrosis is affected by iron protein abnormalities. Hepcidin is a repressor of liver fibrosis. Hepcidin inhibited TGFβ1-induced Smad3 phosphorylation in hepatic stellate cells, which was induced by the action of Akt initiated by a deficiency of ferroportin [264]. NTBI is also responsible for liver fibrosis. Transferrin-KO mice developed liver fibrosis after a high iron diet. The administration of the ferroptosis inhibitor ferrostatin-1 inhibited liver fibrosis [265]. Interestingly, the liver manganese transporter Slc39a14 [266] can transport iron into hepatocytes in the absence of transferrin. Deletion of Slc39a14 in Tf-KO mice also reduced liver fibrosis. The functional role of Slc39a14 in experimental hemochromatosis requires further research.

Ferroportin disease (FD) and hemochromatosis type 4B are characterized by iron overload caused by either reduced cellular iron export in FD or resistance against hepcidin inhibition of ferroportin in HH4B.

As mentioned before, loss-of-function mutations of the SLC40A1 gene lead to iron loading of Kupffer cells and tissue macrophages and decreased iron delivery to Tf causing a low transferrin saturation relative to iron load. Gain-of-function mutations on the other hand, make ferroportin resistant to hepcidin leading to a similar impairment of the hepcidin–ferroportin axis, as found in type 1-3 HH, with high transferrin saturation, and iron loading of hepatocytes [267,268,269].

The R178Q mutation of the *SLC40A1* gene has been described in FD patients from France and Greece [270,271,272]. Recently, 22 patients with a similar mutation from six unrelated families in France, Belgium and Iraq were reported. They showed a reduced iron export capacity and high serum hepcidin concentration [273].

The molecular background of ferroportin disease is not clear. The development of iron overload in patients with reduced iron absorption is contradictory. A possible explanation is that in FD, FPN1 can still traffic to the cell membrane and export iron in enterocytes, but is unable to do so in Kupffer cells and macrophages indicating a different behavior of FPN1 activity between enterocytes and macrophages [269,274]. However, it seems that the pathophysiology of FD and HH4B are not that easy to be explained. In some FD mutations, an element of hepcidin resistance is present indicating that the distinction between the two entities of FPN mutations may not be clear in their pathophysiology [275].

### 3.2. A Proposed New Hypothesis

Summarizing the evidence on the current pathogenesis of HH, certain points raise questions. Thus, the paradox of constant iron deficiency of Kupffer cells and macrophages despite an increased body iron load in all forms of HH (with the exception of FD) was identified many years ago [276]. Constant iron deficiency was also identified in duodenal enterocytes from patients with HH [277].

The current pathogenesis justifies this paradox as a primary reduction of hepcidin leading to FPN1 upregulation in enterocytes and Kupffer cells as detailed above. However, a different behavior of FPN1 has been described at least in FD [274].

Moreover, in β-thalassemia, low hepcidin is also a prominent feature. Then, why are Kupffer cells full of iron? Is Kupffer cell behavior different in thalassemia compared with HH [278]? Repeated transfusions cannot be the explanation, as protracted low hepcidin should remove the extra iron from macrophages.

Hepatocytes also express ferroportin [133,279], but more importantly no different behavior between FPNs of Kupffer cells and hepatocytes has been reported. It is only logical therefore, that in cases of hepcidin deficiency, FPN1 should also be upregulated in hepatocytes leading to iron deficiency rather than iron excess unless there is an unexplained downregulation of FPN1. In mice, data indicate that the classical hepcidin/ferroportin axis is important in the intestine, while the expression of FPN1 in the heart, liver and spleen is mostly regulated by the cellular iron status [280]. If this is true for humans, Kupffer cells and hepatocytes should also behave in a similar way. It has been also shown that macrophage polarization is in part dependent on FPN1 expression. Thus, downregulation of FPN1 in hepatocytes would induce polarization towards the M2 phenotype, while upregulation of FPN1 in hepatocytes as a result of primary reduction of hepcidin should therefore favor M1 polarization [76]. On the other hand, M1 macrophages have an iron storage phenotype, while M2 polarization are not iron storage cells as they have FPN1 upregulation consistent with the status of Kupffer cells in HH as proposed by the current pathogenetic model. This is not consistent with the previous report [70].

To explain the obvious discrepancy between iron storage in hepatocytes and iron deficient enterocytes and Kupffer cells, we propose that HFE-related HH is a primary disorder of Kupffer cells and hepcidin deficiency is a secondary phenomenon. We suggest that Kupffer cells mastermind iron regulation by an unknown factor X signaling to LSECs and enterocytes.
1.**Evidence that Kupffer cells are the critical cell in HH**

Earlier studies showed that in the liver, HFE protein was present on Kupffer cells and LSECs. Kupffer cells from an untreated C282Y HH patient failed to stain with an HFE specific antibody suggesting a primary problem in HH macrophages [281]. The aggregation of iron delivered by holo-Tf was reduced in macrophages from patients with HH compared to controls expressing wild-type HFE. It was suggested, therefore, that the iron deficiency of HH macrophages was a primary effect of the HFE mutation in macrophages [282]. This could be attributed to the strong expression of HFE in macrophages without a similar expression of beta2M, resulting in a decrease in TfR1-dependent iron uptake as has been shown for enterocytes [283].

Experimental evidence indicated that iron efflux from macrophages was inhibited by HFE. Moreover, the HH-associated mutant H41D was unable to inhibit iron release suggesting that HFE has two functions, either binding to TfR1 in collaboration with Tfr, or inhibition of iron release [284]. This was recently confirmed. Selective HFE deletion of myeloid cells positively regulated FPN1 and prevented iron accumulation in macrophages [285]. Macrophages from Hfe−/− mice responded with a reduced inflammatory response to LPS and salmonella challenge also indicating a primary macrophage dysregulation [286].
2.**Indications that low hepcidin in hepatocytes is a secondary phenomenon in HH**

An earlier study suggested that primary hepcidin deficiency as a cause of HH seems unlikely based on the iron liver profile and the hepcidin measurements of a patient transplanted with a hemochromatotic liver [287].

There is recent evidence from hepatocellular cell lines, but also from primary hepatocytes, that hepatocellular iron overload suppresses hepcidin by inhibiting the SMAD signaling pathways downstream of its ligands. Therefore, it may be that hepcidin reduction in HH is secondary to hepatocyte iron overload [156,288].

This is questionable in another study that suggested that elevated intracellular iron has a limited role in hepcidin secretion. However, this study was performed in HepG2 hepatocellular carcinoma cells and not in primary hepatocytes [289]. Interestingly, the inhibitors of BMP pathway, Smad6 and Smad7, were reported to be increased in HFE-HH in accordance with the suggestion that it is the signaling downstream of BMP that is responsible for the hepcidin suppression in HFE-HH. Alternatively, the effect of the unknown factor X from Kupffer cells in the expression of Smad6 and Smad7 cannot be excluded [258].
3.**The Transplantation Experience**

An iron-loaded liver from a patient with occult haemochromatosis was transplanted into a 19-year-old woman. Liver iron was gradually returned to normal [290]. This was not the case when transplantation of a hemochromatotic liver in a patient with primary biliary cholangitis was followed by heavy iron overload up to four years after transplantation. However, consecutive liver biopsies showed that Kupffer cells were also iron loaded indicating that the donor probably suffered from FD type 4A [291]. However, similar results were reported in other cases of inadvertent transplantation of a homozygous for C282Y liver to HFE normal recipients [292,293]. The inadvertent transplantation of a liver from a donor with C282Y/H63D compound heterozygosity into a non-hemochromatotic recipient also resulted in iron overload which occurred over 1.5 years [294].

However, in a larger series, transplantation of iron-loaded liver in normal recipients showed a slow but significant reduction of iron markers including transferrin saturation [295]. Three C282Y heterozygous livers were transplanted in three C282Y heterozygous recipients without development of iron overload. Of course, Kupffer cells are still heterozygous after replenishment [296].

Transplantation of normal livers to HH recipients showed that there was no significant iron overload after liver transplantation [297,298]. These contradictory results may be explained by the fact that the recipient’s Kupffer cells replenish the graft liver over time after transplantation. However, the exact number of new recipient-derived Kupffer cells extensively varies over time [299].

The significance of a primary defect of Kupffer cells was shown by the experiments of liver transplantation in HFE^−/−^ and HFE^+/+^ mice. Transplantation of livers from normal mice in Hfe^−/−^ mice restored the iron-loading phenotype irrespective of HFE expression in enterocytes. However, Kupffer cells remained iron deficient despite liver hepcidin upregulation. This was attributed to Kupffer cell replenishment from the initial recipient progenitors but again the degree of chimerism may influence the results [300,301].

Alternatively, HFE might affect the iron status of macrophages directly rather than acting through hepcidin. It was suggested that macrophages regulate ferroportin degradation by sensing cytoplasmic iron by an HFE-dependent pathway and hepcidin independent mechanism. This may provide an explanation why HFE^−/−^ macrophages, after transplantation into a wild-type recipient, maintain their iron-poor phenotype although serum hepcidin levels are normal [302].
4.**Is There an Indication that Kupffer Cells May Regulate Hepcidin?**

The in vivo depletion of Kupffer cells led to a significant increase in liver hepcidin expression. It was suggested that Kupffer cells control body iron regulation through negative unknown hepcidin-suppressing molecules. We propose instead that it is the limited production of an unidentified positive inducer of hepcidin [303].

This has been shown in co-cultures of HepG2 hepatoma cells with Tamm–Horsfall protein 1 (THP1) macrophages. Upregulation of *HAMP* was induced in the presence of holo-Tf. It was suggested that humoral factors secreted by macrophages are responsible for the induction of hepcidin production [304].

Moreover, there is evidence that the iron sensors from non-parenchymal cells are responsible for BMP6 production. Recent evidence clearly demonstrates that the iron in LSECs is not the decisive factor for BMPs production [305,306]. Incubation of LSECs with iron did not significantly increase the expression of BMP6. On the contrary, treatment of these cells with the iron chelator 2,2′-dipyridyl (2DP) unexpectedly upregulated BMP6 expression suggesting a non-iron-regulation of BMP6 expression in LSECs [307]. It was recently proposed that intracellular aggregation of iron is not essential in either hepatocytes or LSECs for increased BMP6 expression in response to iron overload [308].

The above findings taken together point to the Kupffer cells as the producers of the unidentified iron sensitive signal to control hepcidin.

A different approach was presented by isolation of wild-type bone marrow macrophages and macrophages from H67D HFE knock-in mice (the human equivalent of the H63D variant). Increased hepcidin did not inhibit the release of iron from macrophages initiated by apo-transferrin irrespective of genotype. The demonstration that apo-transferrin increases the release of iron from macrophages indicates that hepcidin is not the only mechanism of iron export by macrophages [309].

There is also evidence of communication between macrophages and hepatocytes in the regulation of hepcidin. Macrophages are able to initiate hepatocyte hepcidin synthesis through secretion of IL-1β and IL-6. Both cytokines are potent inducers of hepcidin as mentioned before. Therefore, a primary Kupffer cell abnormality would reduce hepatocyte hepcidin production [310,311,312] through decreased response to inflammatory signals.

It has been also demonstrated that conventional dendritic cells produce hepcidin that promotes intestinal mucosal healing [313]. Autonomous local hepcidin production by Kupffer cells has not been investigated although immunohistochemistry has provided some indication that this might be the case [314]. In theory, inadequate production may aggravate iron deposition in hepatocytes. Moreover, inadequate production of hepcidin by local macrophages may be responsible for iron deposition in other organs in HH. In myocardial infarction, deletion of hepcidin in macrophages stimulates cardiomyocyte renewal [315].

Secretion of iron-containing hepatocyte extracellular vesicles (EVs) are normally cleared by Kupffer cells. Primarily defective Kupffer will direct hepatocyte iron EVs into hepatic stellate cells, where iron-dependent overproduction of ROS will promote HSC activation towards myofibroblasts and induction of liver fibrosis [316].

Interestingly, recent evidence indicates that activation of mTORC1 in duodenal macrophages initiates the expression of a protease, which degrade transferrin to inhibit iron export from enterocytes [317,318].

Finally, Kupffer cells produce a critical mediator of iron regulation. The neutrophil cytosolic factor 1 (NCF1) triggers oxidation of phospholipids in Kupffer cells promoting TLR4-dependent production of hepcidin by hepatocytes and KC iron overload. Therefore, it is reasonable to hypothesize that KC insufficiency due to mutated HFE will lead to the opposite effect of reduced hepcidin synthesis and reduced iron in KCs [319].
5.**The New Hypothesis**

Based on the above reports, we propose a new hypothesis for the pathogenesis of HFE-related hemochromatosis. We postulate that the central regulator that masterminds body iron regulation is the Kupffer cell and that HFE-related hemochromatosis is a primary Kupffer cell disease. Kupffer cells express high levels of HFE and produce hepcidin. Therefore, they express the HFE-BMP dependent system. We also hypothesize that Kupffer cells produce an unidentified factor X that regulates iron homeostasis by several discrete mechanisms. Factor X promotes the production of BMP6 by LSECs. Furthermore, it increases hepcidin production by acting downstream of the BMP complex and by reducing the expression of the inhibitory SMADs. Finally, factor X increases mTORC1 expression in duodenal macrophages inhibiting iron export from enterocytes. Taken together, reduced production of Factor X by deranged macrophages will lead to the downregulation of hepcidin. Mutations in any of the components of the BMP system and particularly of HFE may reduce the production of Factor X. Therefore, reduced hepcidin and all its consequences are a secondary phenomenon. Moreover, deranged macrophages may produce decreased amounts of IL-6 and IL-1β in response to inflammation and decreased NCF1 reducing the TLR4-mediated production of hepcidin by hepatocytes. These will further reduce the production of hepcidin. Figure 5 is a graphical summary of the above hypothesis.

## 4. Conclusions

Several aspects of body iron homeostasis have not been clarified so far. Hepcidin regulation is a critical factor in iron regulation. The main producer of hepcidin is the hepatocyte, but macrophages are also capable to produce hepcidin. A complex system of bone morphogenic proteins produced by liver sinusoidal endothelial cells are collaborating with proteins involved in the pathogenesis of hereditary hemochromatosis. Factors produced during the process of erythropoiesis and inflammatory cytokines such as IL-6 and IL-1β are also implicated in the regulation of hepcidin. Moreover, many new factors and inhibitors were recently identified. The hereditary hemochromatosis is an example of iron dysregulation. Considerable research identified both the genetic background and aspects of pathophysiology. However, several findings are based on experimental models that may not be relevant to the actual human disease. Nonetheless, the basic pathophysiological abnormality is considered to be a relative deficiency of hepcidin and its consequences. However, there are many points that require clarification. Based on earlier and recent data, we propose a new hypothesis. The main suggestion is that the mastermind regulator of iron homeostasis is the Kupffer cell that comprises almost 80% of tissue macrophages. They control iron regulation by producing an unidentified factor that increases hepcidin production. Therefore, HFE-related hemochromatosis type 1 could be a primary Kupffer cell disease due to the reduced production of this factor and secondary hepcidin downregulation.

## Figures and Tables

**Figure 1 biomedicines-13-00683-f001:**
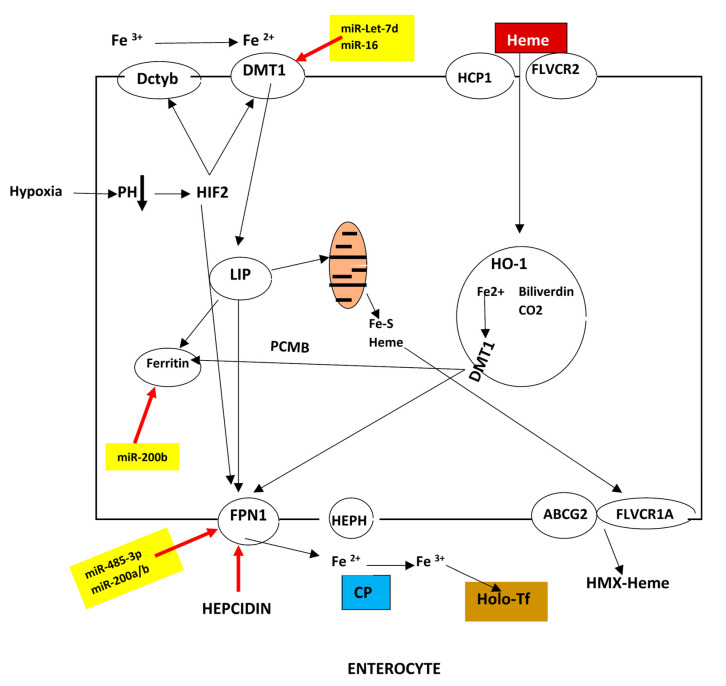
Cellular iron absorption and transport (see also Figure 2). Enterocytes absorb iron from the diet, as either inorganic ferric iron or iron found in heme. The first step is the reduction of ferric iron to the ferrous form by the ferric reductase Dcytb followed by transportation into the enterocyte by DMT1 of the enterocyte apex. In the cytoplasm, iron can be used by mitochondria, can be stored in ferritin or can be exported by ferroportin of the basolateral membrane. The exported iron is oxidized by hephestin and ceruloplasmin to the ferric form and transported by transferrin. Heme is transported through HCP1 and possibly by FLVCR2 [30] and either released through ABCG2 or degraded in endosomes by heme oxygenase. Endogenous heme is also released by FLVCR1A. Hypoxia inactivates prolyl hydroxylases leading to stabilization of HIF2, which promotes the activity of ferroportin, Dctyb and DMT1. On the contrary, certain miRs downregulate ferroportin and DMT1. Black arrows indicate induction. Red arrows indicate inhibition.

**Figure 3 biomedicines-13-00683-f003:**
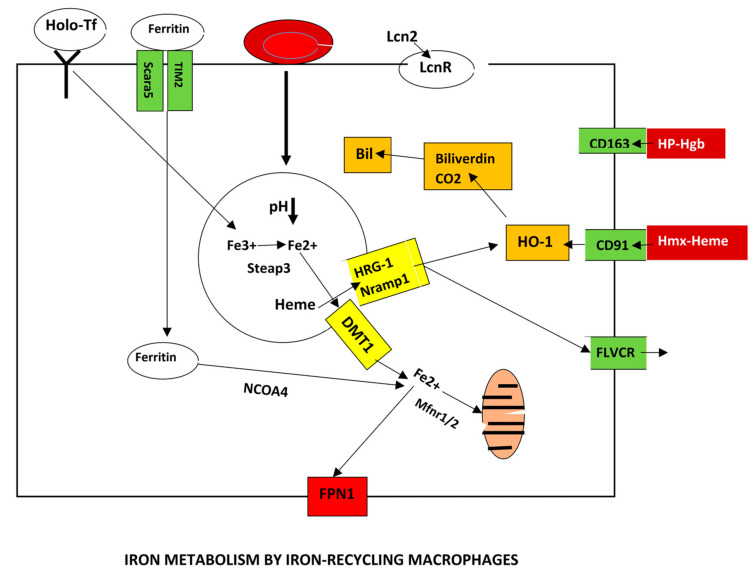
Iron metabolism by iron-recycling macrophages. The main source of iron comes from phagocytosis of aging or damaged erythrocytes that are transported to endosomes. Macrophages also express several receptors to acquire additional iron. Apart from transferrin-bound iron, macrophages can acquire non-transferrin iron through the CD91 and CD163 receptors that scavenge hemopexin-bound heme (Hpx-Heme), and haptoglobin–hemoglobin (Hp-Hgb), respectively. Lipocalin 2 is also taken up by its receptor LcnR. In accordance with other cells, free iron can be transported to mitochondria by Mitoferrin 1/2 (Mfrn1/2) or can be stored in ferritin and exported by ferroportin. Ferritin can also be a source of iron in macrophages. It is taken up by the receptors Scara 5 and Tim2. Endosomal iron is transported to the cytosol by DMT1, while heme is transported by Nramp1 and HRG1 where it is metabolized by HO-1 or is exported via FLVCR.

**Figure 4 biomedicines-13-00683-f004:**
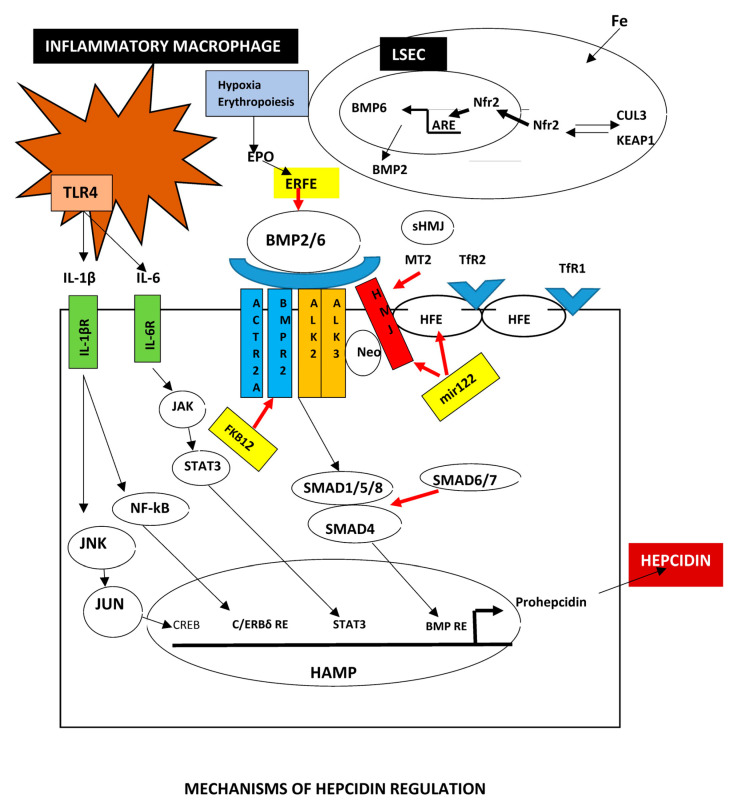
Mechanisms of hepcidin regulation in hepatocytes. Iron excess is taken up by LSECs by an unknown mechanism and initiates the production and release of BMP6. Dissociation of Nrf2 from Keap 1 also induces BMP6 production. BMP2 is constitutively expressed. BMP6, BMP2 and the heterodimer BMP6/BMP2 bind to types I and II BMP receptors activating the BMP/SMAD signaling pathway. Phosphorylation of the SMAD 1/5/8 complex assisted by SMAD4 leads to the translocation into the nucleus and attachment to a BMP-response element (BMP-RE) in the *HAMP* gene encoding hepcidin. In iron overload, holo-Tf binds to TfR1 and TfR2 on the surface of hepatocytes. It is believed that HFE dissociates from TFR1 and attaches to TfR2 and HJV, activating the BMP/SMAD signaling pathway. Neogenin (Neo) stabilizes HJV and the BMP receptor complex. On the contrary, in iron deficiency, TfR1 increases iron uptake from Tf, but also detaches HFE from TfR2 and blocks the BMP/SMAD cascade. SMAD 6 and 7, the immunophilin FKB12, erythroferrone, miR122 and matriptase 2 (MT 2) that cleaves HJV are negative regulators of hepcidin production. Inflammation also may increase hepcidin production by a BMP-independent mechanism. Secretion of IL-6 and IL-1β by macrophages IL-6 activate hepcidin via the JAK/STAT3 and the NF-κB and JNK signaling pathways, respectively, leading to the binding at the relative response elements. For more details, see text. Black arrows indicate induction. Red arrows indicate inhibition.

**Figure 5 biomedicines-13-00683-f005:**
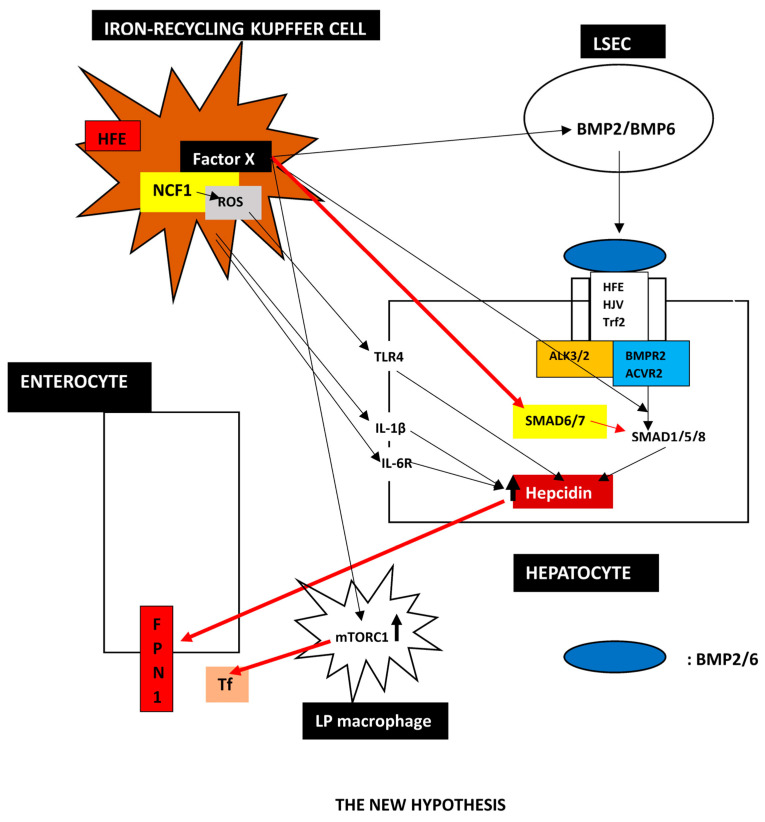
The new hypothesis. For details, see text. Black arrows indicate induction. Red arrows indicate inhibition.

**Table 1 biomedicines-13-00683-t001:** (Modified from Anderson et al. 2021, Alvarenga et al. 2022, Girelli et al. 2022, Hernandez et al. 2021) [236,250,251,252].

HFE-related	p.Cys282Tyr homozygosity or compound heterozygosity of p.Cys282Tyr with other rare HFE pathogenic variants or HFE deletion.
Non-*HFE*-related	Rare pathogenic variants in non-*HFE* genes: -*HJV*-related; -*HAMP*-related; -*TfR2*-related; -*SLC40A1* (very rare gain-of-function variant)-related.
Digenic	Double heterozygosity and/or double homozygosity/heterozygosity for variants in two different genes involved in iron metabolism (HFE and/or non-HFE).
Molecularly undefined	Molecular characterization still not available after sequencing of known genes (provisional diagnosis).

## Data Availability

Not applicable.

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
