# Peer review of "HFE-Related Hemochromatosis May Be a Primary Kupffer Cell Disease"

_biomedicines, 2025, doi:10.3390/biomedicines13030683_

Round 1
Reviewer 1 Report
Comments and Suggestions for Authors
The manuscript presents a new theory regarding the pathophysiology of iron regulation. In order to do so, it extensively reviews the literature on hemochromatosis and hepcidin. However, the theory is by no means supported by the presented facts and the manuscript is far too long.
Major general points:
The title is misleading. The whole purpose of the manuscript is to suggest that hemochromatosis is primary Kupffer cell disease, and that the well-documented decrease in hepcidin is secondary. However, the term “Hereditary hemochromatosis” includes also juvenile hemochromatosis caused by hepcidin mutations, where decreased hepcidin is undoubtedly the primary cause of the disease. Therefore, the title must be altered.
Gene symbols should be in italics.
The manuscript is so long that it is difficult to read. The authors obviously tried to comment on every known aspect of hepcidin regulation, but this is unnecessary. It is suggested they cite only those papers relevant to their hypothesis, plus the most important papers regarding hepcidin regulation.
The manuscript includes over 300 references. It is therefore confusing that some of the most important references are missing: The key paper on hepcidin regulation by Pigeon et al, the discovery of hepcidin regulation by erythropoietin by Nicolas et al., the discovery of the role of bone morphogenetic protein signaling in hepcidin regulation by Babitt et al, the discovery of the role of BMP6 in hepcidin regulation by Meynard et al, and the discovery of erythroferrone by Kautz and Ganz.
Specific points:
Line 146 mentions the (controversial) export of iron by exosomes, supported by two references. However, reference 33 is a review, not an original paper, while reference 35 does not mention exosomes at all.
Line 181: ALKs are by no means the receptors implicated in iron entry to LSECs.
Line 369: The hierarchy of inflammation versus iron deficiency has been clearly addressed in mouse experiments by Constante and Santos in 2006. In their experiments, the inflammatory pathway clearly overrides iron deficiency. Reference 115 refers to MILD inflammation.
Line 660: EPO does not “target” ERFE, it induces its expression.
Line 674: In vivo FGL1 induction by hypoxia has not been demonstrated.
Line 821: In HH type 1 iron load is NOT COMMON? An extremely misleading statement. Of course, only a fraction of patients with mutated HFE develop iron overload, but this does not mean that iron overload is absent in patients with clinically manifest hemochromatosis.
Line 938: The question asked here is confusing. Thalassemia typically requires repeated transfusions, these would be an obvious explanation for macrophage iron loading.
Line 943: Berezovsky et al have shown in 2022 that iron overload increases liver FPN protein in mice despite the increase in hepcidin. Obviously, liver FPN protein is primarily regulated by iron, rather than by hepcidin.
Line 978: In vivo, iron overload definitely increases SMAD signaling. Reference 297 refers solely to cell lines and explicitly states so.
Line 1110: The latest sentence should mention that it refers to HH type 1. “Is” should be replaced by “could”, since the presented evidence is by no means compelling.
Minor poits:
Check spelling! For example, lines and 362 and 1051 mention “hepsidin” instead of “hepcidin”. The all important last sentence refers to Hreditary hemochromatosis…
Comments on the Quality of English Language
Carefull spell check is necessary.
Author Response
REVIEWER 1
We thank the reviewer for the comments. All changes are highlighted in yellow color. Please note that due to extensive re-organization of the paper, lines and number of original references do not coincide with those on the final manuscript.
Major general points:
1.The title is misleading. The whole purpose of the manuscript is to suggest that hemochromatosis is primary Kupffer cell disease, and that the well-documented decrease in hepcidin is secondary. However, the term “Hereditary hemochromatosis” includes also juvenile hemochromatosis caused by hepcidin mutations, where decreased hepcidin is undoubtedly the primary cause of the disease. Therefore, the title must be altered.
The title has been altered
2.Gene symbols should be in italics.
It has been done
3.The manuscript is so long that it is difficult to read. The authors obviously tried to comment on every known aspect of hepcidin regulation, but this is unnecessary. It is suggested they cite only those papers relevant to their hypothesis, plus the most important papers regarding hepcidin regulation.
The manuscript has been re-organized and subsections on BMPs and hepcidin have been shortened.
4.The manuscript includes over 300 references. It is therefore confusing that some of the most important references are missing: The key paper on hepcidin regulation by Pigeon et al, the discovery of hepcidin regulation by erythropoietin by Nicolas et al., the discovery of the role of bone morphogenetic protein signaling in hepcidin regulation by Babitt et al, the discovery of the role of BMP6 in hepcidin regulation by Meynard et al, and the discovery of erythroferrone by Kautz and Ganz.
The suggested papers have been included in the relevant sections.
Specific points:
Line 146 mentions the (controversial) export of iron by exosomes, supported by two references. However, reference 33 is a review, not an original paper, while reference 35 does not mention exosomes at all.
Ref 35 has been replaced by two recent original papers.
Line 181: ALKs are by no means the receptors implicated in iron entry to LSECs.
The expression was corrected
Line 369: The hierarchy of inflammation versus iron deficiency has been clearly addressed in mouse experiments by Constante and Santos in 2006. In their experiments, the inflammatory pathway clearly overrides iron deficiency. Reference 115 refers to MILD inflammation.
We included the reference by Constante and commented on that.
Line 660: EPO does not “target” ERFE, it induces its expression.
Corrected
Line 674: In vivo FGL1 induction by hypoxia has not been demonstrated.
We commented on that.
Line 821: In HH type 1 iron load is NOT COMMON? An extremely misleading statement. Of course, only a fraction of patients with mutated HFE develop iron overload, but this does not mean that iron overload is absent in patients with clinically manifest hemochromatosis.
The expression was corrected to be clear.
Line 938: The question asked here is confusing. Thalassemia typically requires repeated transfusions, these would be an obvious explanation for macrophage iron loading.
We commented on that.
Line 943: Berezovsky et al have shown in 2022 that iron overload increases liver FPN protein in mice despite the increase in hepcidin. Obviously, liver FPN protein is primarily regulated by iron, rather than by hepcidin.
The paper by Berezovsky was included. However, the regulation of FPN1 expression by the cellular iron status (if true for humans) does not change the fact that Kupffer cell and hepatocyte FPN1 do not behave similarly.
Line 978: In vivo, iron overload definitely increases SMAD signaling. Reference 297 refers solely to cell lines and explicitly states so.
We pointed out that the evidence comes from hepatocellular cell lines but also from primary hepatocyte cultures (Ref 101).
Line 1110: The latest sentence should mention that it refers to HH type 1. “Is” should be replaced by “could”, since the presented evidence is by no means compelling.
We did that
Minor poits:
Check spelling! For example, lines and 362 and 1051 mention “hepsidin” instead of “hepcidin”. The all important last sentence refers to Hereditary hemochromatosis…
We did that. Spelling, grammar and syntax mistakes have been corrected.
Reviewer 2 Report
Comments and Suggestions for Authors
The manuscript by Elias Kouroumalis, Ioannis Tsomidis, and Argyro Voumvouraki “Hereditary Hemochromatosis May Be a Primary Kupffer Cell Disease” gave a broad overview of iron metabolism and genetic mechanisms of iron overload disease and hereditary hemochromatosis. Authors present the original hypothesis that Kupffer cells are primary regulators and contributors to hemochromatosis by producing unidentified factor X. Authors address essential questions. Still, there are some flaws in the manuscripts that should be addressed. The overall review is poorly structured and too long, and the manuscript has many redundancies, making it hard to comprehend. It would be beneficial to present outlines of the review and make it more concise. The figures are of poor quality and confusing. Also, there are some discrepancies in the references. Here are my additional comments on the manuscript.
Figure 1 . FLVCR1a does not transport dietary heme, only endogenous heme (PMID: 26067085). Alco, FLVCR1A recently was found is also a choline receptor (PMID: 37100056).
Line 112-113
“It should be noted that HO-1 is cytoprotective against cell death, including necrosis, necroptosis, pyroptosis. In ferroptosis, HO-1 may be detrimental as it enhances iron release.”
This statement is too vague. The protective or effect of HO-1 depends on the cellular context and tissue.
Line 116-117 and Figure 1describes the iron absorption by the enterocyte.
The figure presents more broader information on iron and heme transport, dietary and endogenous.
Line 140. “Storage in ferritin demands oxidation of ferrous to ferric iron, which is 140 achieved by the ferroxidase activity of H-ferritin [34] “ Ref 34 is irrelevant to ferroxidase activity of ferritin
- Cohen, L.A.; Gutierrez, L.; Weiss, A.; Leichtmann-Bardoogo, Y.; Zhang, D.L.; Crooks, D.R.; Sougrat, R.; Morgenstern, A.; Galy, B.; Hentze, M.W.; Lazaro, F.J.; Rouault, T.A.; Meyron-Holtz, E.G. Serum ferritin is derived primarily from macrophages through a nonclassical secretory pathway. Blood. 2010, 116, 1574-1584. .
Line 152
“Deficiency of one iron leads to the loading of copper by a still unknown mechanism”. This is unclear sentence.
Line 180
“Iron is captured by unknown receptor(s) on the surface of LSECs leading to increased mitochondrial reactive oxygen species (ROS), which activates Nrf2 to upregulate BMP6 transcription”.
This statement requires a reference (PMID: 31276102)
Line 201
“However, in acute increase of BMP2 –induced hepcidin expression, hemochromatosis-related proteins are not essential for the activation of BMP-SMAD signaling pathway by BMP2 [56]” This sentence is confusing and does not make sentence.
Line 232
“recent review of BMPs and iron homeostasis has been published [51].” This sentence does not make sense, as authors just described paragraph with recent work on papers published in 2024 and reference 51 was published in 2020.
Line 235
The refence 64 is outdated and not relevant to the previous statements.
“64. Koskenkorva-Frank, T.S.; Weiss, G.; Koppenol, WH.; Burckhardt, S. The complex interplay of iron metabolism, reactive oxygen species, and reactive nitrogen species: insights into the potential of various iron therapies to induce oxidative and nitrosative
stress. Free Radic Biol Med. 2013, 65, 1174-1194.”
Line 250
“integrity and phenotype (i.e. fenestrae), iron-induced Nrf2 activity may contribute to endothelial dysfunction [68,69].”
The reference is 69 is not relevant to the statement.
Line 1030
“Moreover, there is evidence that the iron sensors from non-parenchymal cells are responsible for BMP6 production [314].”
This reference to the review of 2015 is outdated.
Line 1034
“of these cells with the iron cheletor 2DP unexpectedly upregulated BMP6 expression suggesting a non.iron-regulation of BMP6 expression in LSECs [317].”
Please give full name of iron chelator mentioned in the reference, 2,2′-dipyridyl.
Author Response
REVIEWER 2
We thank the reviewer for the comments. All changes are highlighted in yellow color. Please note that due to extensive re-organization of the paper, lines and number of original references do not coincide with those on the final manuscript.
The manuscript by Elias Kouroumalis, Ioannis Tsomidis, and Argyro Voumvouraki “Hereditary Hemochromatosis May Be a Primary Kupffer Cell Disease” gave a broad overview of iron metabolism and genetic mechanisms of iron overload disease and hereditary hemochromatosis. Authors present the original hypothesis that Kupffer cells are primary regulators and contributors to hemochromatosis by producing unidentified factor X. Authors address essential questions. Still, there are some flaws in the manuscripts that should be addressed. The overall review is poorly structured and too long, and the manuscript has many redundancies, making it hard to comprehend. It would be beneficial to present outlines of the review and make it more concise. The figures are of poor quality and confusing. Also, there are some discrepancies in the references. Here are my additional comments on the manuscript.
The paper has been extensively re-organized, redundancies have been omitted and subsections have been shortened.
The quality of the figures will be improved in the final manuscript.
Figure 1. FLVCR1a does not transport dietary heme, only endogenous heme (PMID: 26067085). Alco, FLVCR1A recently was found is also a choline receptor (PMID: 37100056).
Fig 1 has been modified to clarify this and a note has been made in the legend.
Line 112-113
“It should be noted that HO-1 is cytoprotective against cell death, including necrosis, necroptosis, pyroptosis. In ferroptosis, HO-1 may be detrimental as it enhances iron release.”
This statement is too vague. The protective or effect of HO-1 depends on the cellular context and tissue.
This has been clarified
Line 116-117 and Figure 1describes the iron absorption by the enterocyte.
The figure presents more broader information on iron and heme transport, dietary and endogenous.
We clarified this point
Line 140. “Storage in ferritin demands oxidation of ferrous to ferric iron, which is 140 achieved by the ferroxidase activity of H-ferritin [34] “ Ref 34 is irrelevant to ferroxidase activity of ferritin
- Cohen, L.A.; Gutierrez, L.; Weiss, A.; Leichtmann-Bardoogo, Y.; Zhang, D.L.; Crooks, D.R.; Sougrat, R.; Morgenstern, A.; Galy, B.; Hentze, M.W.; Lazaro, F.J.; Rouault, T.A.; Meyron-Holtz, E.G. Serum ferritin is derived primarily from macrophages through a nonclassical secretory pathway. Blood. 2010, 116, 1574-1584. .
Ref 34 was misplaced and was placed in the right position under a new enumeration due to manuscript changes. Ferritin ferroxidase was supported by the correct references.
Line 152
“Deficiency of one iron leads to the loading of copper by a still unknown mechanism”. This is unclear sentence.
Sentence was corrected by omitting the unnecessary word “one”
Line 180
“Iron is captured by unknown receptor(s) on the surface of LSECs leading to increased mitochondrial reactive oxygen species (ROS), which activates Nrf2 to upregulate BMP6 transcription”.
This statement requires a reference (PMID: 31276102)
Reference has been added.
Line 201
“However, in acute increase of BMP2 –induced hepcidin expression, hemochromatosis-related proteins are not essential for the activation of BMP-SMAD signaling pathway by BMP2 [56]” This sentence is confusing and does not make sentence.
The sentence has been completed and rephrased.
Line 232
“recent review of BMPs and iron homeostasis has been published [51].” This sentence does not make sense, as authors just described paragraph with recent work on papers published in 2024 and reference 51 was published in 2020.
Sentence has been removed.
Line 235
The refence 64 is outdated and not relevant to the previous statements.
“64. Koskenkorva-Frank, T.S.; Weiss, G.; Koppenol, WH.; Burckhardt, S. The complex interplay of iron metabolism, reactive oxygen species, and reactive nitrogen species: insights into the potential of various iron therapies to induce oxidative and nitrosative
stress. Free Radic Biol Med. 2013, 65, 1174-1194.”
Ref 64 has been removed
Line 250
“integrity and phenotype (i.e. fenestrae), iron-induced Nrf2 activity may contribute to endothelial dysfunction [68,69].”
The reference is 69 is not relevant to the statement.
Ref 68 and 69 have been re-allocated under new numbering to clarify the statement made.
Line 1030
“Moreover, there is evidence that the iron sensors from non-parenchymal cells are responsible for BMP6 production [314].”
This reference to the review of 2015 is outdated
Ref 314 has been removed.
Line 1034
“of these cells with the iron cheletor 2DP unexpectedly upregulated BMP6 expression suggesting a non.iron-regulation of BMP6 expression in LSECs [317].”
Please give full name of iron chelator mentioned in the reference, 2,2′-dipyridyl.
It was done
Spelling, grammar and syntax mistakes have been corrected.
Round 2
Reviewer 2 Report
Comments and Suggestions for Authors
The authors answered my questions, addressed the comments, and changed the text and figures. I recommend publishing this manuscript.